



# Impact of drought hazards on flow regimes in anthropogenically impacted streams: an isotopic perspective on climate stress

Maria Magdalena Warter[1], Dörthe Tetzlaff [1,2,4], Christian Marx[3] and Chris Soulsby[1,3,4]

[1]Department for Ecohydrology & Biogeochemistry, Leibniz Institute of Freshwater Ecology and Inland Fisheries (IGB), Berlin, Germany

[2] Department of Geography, Humboldt University Berlin, Berlin, Germany

[3] Chair Water Resources Management and Modeling of Hydrosystems, Technische Universität Berlin, Berlin, Germany

[4] Northern Rivers Institute, School of Geosciences, University of Aberdeen, Aberdeen, UK

*Correspondence to*: Dr. Maria Magdalena Warter (maria.warter@igb-berlin.de)

**Abstract**

Flow regimes are increasingly impacted by more extreme natural hazards of droughts and floods as a result of climate change, compounded by anthropogenic influences in both urban and intensively managed rural catchments. However, the characteristics of sustainable flow regimes that are needed to maintain or restore hydrologic, biogeochemical and ecological function under rapid global change remain unclear and contested. We conducted an inter-comparison of two streams in the Berlin-Brandenburg region of NE Germany, which are both mesoscale sub-catchments of the River Spree; an intermittent rural agricultural stream (the Demnitzer Millcreek) and a heavily anthropogenically impacted urban stream (the Panke). Through tracer-based analyses using stable water isotopes, we identified the dominant physical processes (runoff sources, flowpaths and age characteristics) sustaining streamflow over multiple years (2018-2023), including three major drought years (2018-20, 2021-22). In the urban stream, low flows are regulated through artificially increased baseflow from treated waste water effluent (by up to 80%), whilst storm drainage drives rapid, transient high flow and runoff responses (up to 80%) to intense convective summer rainfall. The intermittent groundwater-dominated rural stream experienced extended no-flow periods during drought years (~ 60% of the year), and only moderate storm runoff coefficients (<10%) in winter along near-surface flows paths after heavy rainfall. In both streams, groundwater dominance with young water influence prevails, with low water ages in the urban stream (<10%) despite significant urban runoff, and higher ones in the rural stream (~15%). Urban cover resulted in mean transit time of ~4 years compared to arable land with ~3 years, highlighting the interlinkages of landuse and catchment properties on catchment transit times. Understanding seasonal and interannual variability in streamflow generation through a tracer-based hydrological template, has potential for assessing the impacts of natural hazards on the sustainability of future baseflow management, including wider water quality and ecological implications across anthropogenically impacted environments.



## 1 Introduction

Urbanization and anthropogenic alterations to hydrological pathways, drainage networks and flow regimes have progressively changed the water balance and dynamics of contemporary streams and rivers, increasing their sensitivity and impacts to climatological and hydro-meteorological hazards (Bonneau et al., 2018; Soulsby et al., 2014; Stewardson et al., 2017). Along with the well-established impacts of anthropic changes imposed on urban freshwaters, many other areas, including peri-urban and rural agricultural environments, are also experiencing dramatic alterations to natural flow regimes and hydrologic processes (Döll & Schmied, 2012; Yang et al., 2011). These changes are propagated by the persistent reorganization of surface and subsurface hydrological flowpaths, widespread landuse changes and stream network alterations, as well as increasing baseflow manipulations (Bonneau et al., 2018; Marx et al., 2021; Oswald et al., 2023; Soulsby et al., 2014).

In recent years the frequency and intensity of hazards such as floods and severe multi-year droughts have contributed to the proliferation of further hydrological changes, which have given rise to a paradigm shift in future streamflow management needs and recognition of a persistent lack of understanding of essential hydrologic processes in urban and other anthropogenically impacted systems (Arthington et al., 2006; Oswald et al., 2023). Despite the importance of natural flow variability (Poff et al., 1997; Stewardson et al., 2017) and numerous studies demonstrating the effects of changes in natural flow regimes on hydrological and ecological function (Arthington et al., 2006; Bhaskar et al., 2016; Olden & Poff, 2003; Poff & Zimmerman, 2010; Tetzlaff et al., 2005), there is still a distinct lack of agreement on how to understand and manage flow regimes and their evolution in the face of rapid global change. This provides a weak evidence base for managers wanting to maintain or restore a baseline of natural flow regime characteristics that supports the hydrologic, biogeochemical and ecological functionality of freshwater systems that provide important ecosystem services (Acreman et al., 2014; Arthington et al., 2006).

Concerns over water stress and drought as drivers of rapid hydrological change have intensified both in cities (Kuhlemann et al., 2020; Paton et al., 2021), and lowland agricultural catchments with intermittent stream systems (Kleine, Tetzlaff, et al., 2021; O'Briain, 2019; Wu et al., 2021). During recent severe drought years (2018-2020) in northern and central Europe, significant shifts of streamflow from perennial to intermittent were widely observed, with the probability and longevity of intermittency likely to increase further with projected increases of temperatures across Europe (Kleine et al., 2020; Lobanova et al., 2018; Sarremejane et al., 2022; Tramblay et al., 2021). In addition, ongoing urban densification contributes to increasingly flashy hydrographs, deteriorating water quality and increased influence of waste water discharges, causing flow regimes to increasingly deviate from the "natural flow paradigm" with a seasonal succession of high and low flows (Bhaskar et al., 2016; Bonneau et al., 2018; Marx et al., 2021; Soulsby et al., 2015).

Projected changes to hydroclimate and associated changes in hydrological partitioning (e.g. runoff, artificial drainage, recharge and evapotranspiration) create challenges for water resource management, to anticipate future resilience of natural





flow regimes and maintain stream biodiversity and ecological integrity (Acreman et al., 2014; Tonkin et al., 2021). Biodiversity and the health of aquatic and wetland ecosystems hinge on ecological processes are dependent on the natural occurrence and variability of high and low flows: In particular, baseflows exert critical controls on habitat maintenance and the survival of

different aquatic species, as well as the moderation of water temperatures, water quality, oxygen levels, nutrient loads and vegetation growth (Arthington et al., 2006; Poff & Zimmerman, 2010; Stewardson et al., 2017).Assessment of future hydroclimate variability and its effects on streamflow generation in anthropogenically impacted systems, calls for a process-based understanding of event-based, seasonal and interannual streamflow patterns and their interactions with climate and the landscape (Kleine et al., 2020; Tetzlaff et al., 2005).

A landscape scale understanding of controls on streamflow regimes requires an integrative approach that captures the ecologically and hydrological meaningful characteristics of seasonal flow dynamics (Arthington et al., 2006; Tetzlaff et al., 2005; Tonkin et al., 2021). Environmental tracers, such as stable water isotopes, can be useful for characterizing complex hydrological systems in order to understand hydrological functioning across multiple scales (Ehleringer et al., 2016; Jasechko, 2019; Kendall & McDonnell, 1998; Stevenson et al., 2022). Tracer applications and tracer-based models can provide insight

into controls on streamflow generation across different climatic and geographic scales (Bonneau et al., 2018; Stevenson et al., 2022; Von Freyberg et al., 2018). The conservative behaviour of stable water isotope ratios of water ($\delta^{18}$O, $\delta^{2}$H) and their ability to integrate hydrological processes make them useful indicators of water sources and flowpaths (Ehleringer et al., 2016; Marx et al., 2021; Von Freyberg et al., 2018). This can help quantify drought effects (Kleine et al., 2020; Kuhlemann et al., 2020; Smith et al., 2020), mean transit times and water ages (Birkel et al., 2016; Hrachowitz et al., 2010; Soulsby et al., 2015;

Tetzlaff, Buttle, Carey, Mcguire, et al., 2015), as well as groundwater-surface water interactions and recharge (Wallace et al., 2021; Ying et al., 2024) across a range of temporal and spatial scales.

We conducted an inter-comparison between an urbanized and a rural agricultural catchment in the Berlin-Brandenburg region, which experienced different extreme hydroclimatic conditions over a five year period, including a severe drought period between 2018-2020 (Creutzfeldt et al., 2021). Such assessments can be especially useful for as part of inter-catchment

comparisons (Tetzlaff et al., 2015). Contemporary tracer-based catchment inter-comparisons using high resolution isotope data are still rare in the context of urbanization and agricultural intensification (Bonneau et al., 2018; Marx et al., 2023). Despite a wet year in 2017, 2018 had a dry late winter with low regional groundwater levels, which progressively declined during a hot, dry summer, turning into an extreme 200 year drought with significant impacts for agriculture, ecology and river flows (Beillouin et al., 2020; Rousi et al., 2023). Under the most climate change scenarios, further declines in groundwater levels by

the end of the century are expected for North and Eastern Germany with dry-hot seasons getting longer, exacerbating existing patterns of increased streamflow intermittency and low flows (Tramblay et al., 2021; Wunsch et al., 2022). Therefore, we sought to understand how drought and intensive storm events have influenced the hydrological and physical functioning of streamflow regimes in two contrasting anthropogenically impacted catchments.



Using stable water isotopes as tracers, we specifically aim to assess i) how drought affects metrics of hydrological functioning such as water partitioning, runoff sources, transit times and water ages in two contrasting catchments through an inter-comparison, ii) characterize streamflow persistence and resilience during drought periods and in response to select intensive storm events as exemplary hazards and iii) understand implications for changing flow regimes under projected future hydroclimate perturbations. We focused on characterizing the contrasting streamflow responses between urban systems with large urban storm drain effects and artificially increased baseflow, vs rural groundwater-fed intermittent systems with agricultural drainage. in an integrated way. In so doing, we sought to provide the hydrological context and evidence basis to help environmental decision makers to establish sustainable environmental flow targets and evaluate the sensitivity of the natural flow regime baseline likely required to maintain biodiversity and ecological integrity (Poff & Zimmerman, 2010; Tonkin et al., 2021).

## 2 Study Catchments

Demnitz Millcreek - a rural agriculturally influenced stream in the State of Brandenburg, and the River Panke – a heavily urbanized stream in Berlin, were compared (Fig. 1). Both catchments (hereafter referred to as the rural and urban catchments) are situated in one of the driest parts of NE Germany, with 577 mm annual precipitation (1981-2010 average), distributed throughout the year as frequent, lower-intensity frontal winter rains and infrequent heavy convective storms in summer (DWD, 2023). However, regional climate differences exist between dense urban Berlin and the drier lowland Brandenburg region, where potential evapotranspiration generally exceeds annual precipitation inputs (~700mm/yr) (DWD, 2023).

The rural catchment (~66 km$^2$) is a tributary of the River Spree, influenced by the regional groundwater system of the Spree valley. The catchment has a mixed landuse, including forest, wetland, cereal crops and pastures, with only small, distributed villages giving 2% urbanized area (Fig. 1b). A full table of landuse distribution is provided in Supplementary Materials (Table S1). The site is part of a long-term ecohydrological observatory and extensive meteorological and ecohydrological data has been measured at multiple temporal and spatial scales since 2018 (Tetzlaff et al., 2023). Arable non-irrigated landuse is highest in the northern part of the catchment (~68%). Agricultural crops primarily include water-demanding cereal crops, such as winter wheat, barley, and maize, occupying the higher quality soils, whereas in the lower part of the catchment mixed forest and wetlands dominate, which are primarily used as pastureland (Fig. 1d) (Smith et al., 2021).

The flat lowland landscape reflects a strong influence of glaciation, resulting in characteristics sections of unmixed sediments. While soils in the agricultural areas comprise silty brown-earths, the lower catchment is dominated by sandy soils with low water retention. The river channel network is fringed by peaty soil, particularly in the central wetlands (Fig 1f). The upper catchment is underlain by unconsolidated sediments of a ground moraine, with a moderately permeable unconfined aquifer and shallow groundwater levels within a few meters of the surface (Fig. 1f) (Ying et al., 2024). The stream network has been significantly altered by historical agricultural management, resulting in a high density of drainage ditches and artificial channels, affecting nutrient transport and water quality. To increase nutrient retention and regulate water quality, a nature-
based solution approach was adopted by restoring the central wetlands by partially blocking drains (starting in 2000), installing a weir and shallowing wetland channels, which increased the water retention in the area and led to a subsequent recolonization by beavers (around 2007) (Smith et al., 2020). The rural catchment is highly drought sensitive, with an intermittent groundwater-dominated flow regime (with the stream being dry most years for 3-4 months) and overall low run-off coefficients

even during wet periods (<10%) due to high evaporative losses (Kleine et al., 2020). Groundwater recharge primarily occurs during the cold/wet season, driving seasonal baseflow onto which streamflow responses to seasonal precipitation are superimposed.

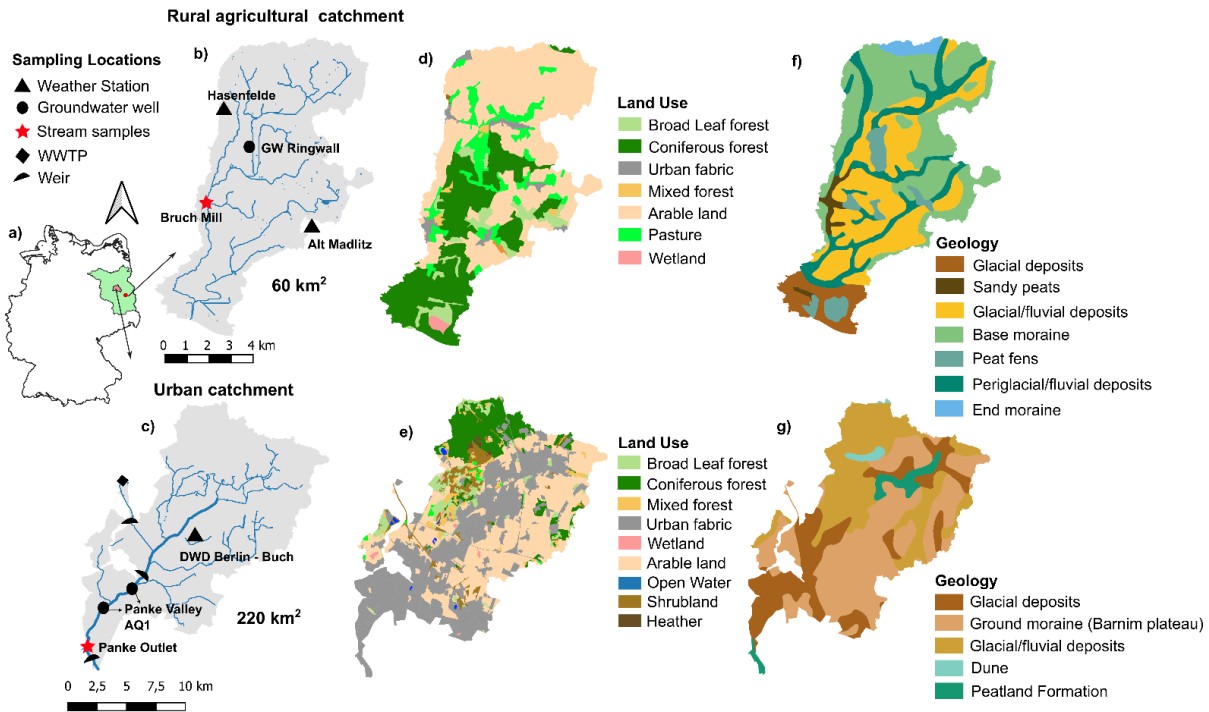

**Figure 1: a) Location of catchments within the Berlin-Brandenburg region. b, c) Overview of sampling locations in the rural and urban catchment respectively, including stream sampling locations (red) of regular stream isotopic samples, groundwater**
**monitoring wells, weather stations and in stream weirs; d, e) Distribution of landuse and f, g) Map of the geology in both catchments. (Basemaps: LGB (Landesvermessung und Geobasis Information Brandenburg) 2020; Umweltatlas Berlin/ALKIS, 2020)**

The urban catchment (~220 km²) is drained by the highly regulated Panke river, also a tributary of the Spree, draining a densely urbanized landscape (Fig. 1c). The headwaters are predominantly rural, lying on the northern edge of the Warsaw-Berlin glacial spillway (Fig. 1g). The main geological unit of the shallow aquifer (AQ1) that feeds the catchment is partially
confined in the East by an overlying ground moraine, and unconfined in the lower catchment with layers of sands and gravels overlying an aquitard of glacial till. Like the rural catchment, the stream network drains silty soils in the headwaters and sandy soils in the south.

In the North, ~22% of the catchment is covered by urban fabric and streamflow is generally groundwater-dominated with seasonally varying inflows from agricultural and forested areas, in addition to the impact of urban storm drains, with some





stretches of the river dry in summer  (Marx et al., 2021) (Fig. 1e). In contrast, the lower catchment is more densely urbanized (~40%), with the stream receiving substantial amounts of urban runoff from impervious surfaces and stormwater drains, as well as large contributions of effluent discharge as treated sewage effluent. Effluent releases generally vary in response to season, with an increased volume of wastewater flowing into the river (up to 80%) during the drier summer periods to enhance baseflow. Conversely, during wetter periods and in case of heavy summer-storm events, baseflow is lowered through weir

operations and storm runoff from the upper catchment is diverted it to the neighbouring Tegeler catchment and the Nordgraben (Marx et al., 2023). This has led to a highly artificial flow regime with no clear seasonal variation between high and low flows. In recent years the catchment has been subject to targeted stream restoration to improve ecological conditions and water quality, as well as flood mitigation by rainwater management (SenUVK, 2009).

## 3 Data and Methods

### 3.1 Climate and hydrological data


In the rural catchment, hourly meteorological data were obtained for the period 2018-2023 from automatic weather stations (AWS) at locations in Hasenfelde (WLV, Environmental Measurement Limited, UK) and Alt-Madlitz (Campbell Scientific, USA) (Fig. 1b). The AWS recorded radiation, air temperature, precipitation, relative humidity and ground heat flux at a 15-min resolution. Discharge was measured at Bruch Mill from water level measurements by pressure sensors (AquiLite ATP 10,

Aquitronic Umweltmeßtechnik GmbH, Kirchheim/Teck, Germany) and an established rating curve (Smith et al., 2020). Groundwater was monitored from a location in a forest in the central catchment (GW Ringwall, screened 2-4 m below surface). Groundwater levels were logged every 4hrs (AquiLite ATP10, AquiTronic Umwelmeßtechnik, GmbH, Kirchheim/Teck, Germany). For the urban catchment, similar hourly meteorological data (2018-2023) were obtained from a climate station (Buch) of the German Weather Service (DWD, 2023) (Fig. 1c). 15-min as well as daily discharge data for a station in the lower

Panke catchment (Bürgerpark) and groundwater levels from wells in the unconfined Panketal aquifer (AQ1) were obtained from the Berlin Senate (SenUVK, 2023).

To assess water fluxes relevant to seasonal streamflow dynamics, hydrological and climate data differentiated into hydrological years (October $1^{st}$ – September $30^{th}$), which were used to calculate different metrics for each stream, such as total runoff ($Q_{Ro}$), storm totals and intensity, storm duration as well as annual and seasonal precipitation totals and magnitudes

(minimum, maximum) of discharge. Discharge was normalized to catchment areas and flow duration curves derived to characterize interannual variability. Hydrograph separation of stream discharge into baseflow ($Q_B$) and stormflow ($Q_S$) was achieved using HydRun, a MATLAB-based toolbox (Tang & Carey, 2017), which is based on a recursive digital filter technique developed by Nathan & McMahon (1990). The filter coefficients (fc) and number of filter passes ranged between 0.7-0.99 and 0-10 respectively. Flashiness, as the rate of change in streamflow, was estimated through the Richard Baker

Flashiness index (Baker et al., 2004), using daily storm flow data.





To examine storm activity and streamflow responses, precipitation data was separated into summer (June-September) and winter data (October -May). Hourly precipitation characterized total event precipitation ($P_{total}$, mm), mean precipitation intensity ($P_{int}$, mm h$^{-1}$), streamflow peak ($Q_{peak}$, ls$^{-1}$km$^2$), maximum precipitation over 1 h ($P_{max}$, mm h$^{-1}$) and rainfall duration (T, hours). Storm events were identified automatically, whereby precipitation events exceeding one hour were aggregated; for multiple consecutive events with <5hr breaks or multi-day events, the event precipitation was summarized into one total storm amount. Statistical differences in storm characteristics were assessed through Kolmogorov-Smirnov tests and Spearman rank correlations, using a p-threshold of at least 0.05 (95% confidence level). The annual runoff coefficient was calculated as the ratio between stormflow and precipitation using annual runoff and total annual precipitation ($Q_{Ro}$/P), while total annual runoff was estimated using annual discharge and baseflow totals ($Q_{Ro} = Q_S – Q_B$). The baseflow index (BFI) was calculated as a ratio of the total baseflow volume to the total runoff volume for each hydrological year, to assess the proportion of stream runoff derived from stored sources (e.g. groundwater).

### 3.2 Stable water isotopes

Daily precipitation isotopes were collected at Hasenfelde (rural) from 2018, and at the Steglitz Urban Ecohydrological Observatory (SUEO) (urban) in SE Berlin from 2019, using a modified ISCO 3700 (Teledyne Isco Lincoln, USA) automated samplers. Samples were protected from evaporation by a paraffin layer (thickness > 0.5mm, IAEA/GNIP precipitation sampling guide V2.02 September 2014). At Bruch Mill (rural), daily stream water isotopes were sampled from 2018, at 16:00 each day, using an automated ISCO 3700 (also protected from evaporation by paraffin). In the urban catchment, daily stable water isotope samples were collected manually from October 2019 until December 2022, and weekly isotopes from 2023 onwards, near the most downstream gauging station (Fig. 1c). Monthly groundwater isotope samples were collected from multiple wells in the AQ1 aquifer as part of a measurement campaign in 2020/2021 (Marx et al., 2021). All samples were filtered (0.2 µm cellulose acetate) into 1.5ml vials and analyzed for $\delta^{18}O$ and $\delta^2H$ using a Picarro L-2130-I cavity ring-down water isotope analyzer (Picarro Inc., Santa Clara, CA, USA) in reference to the Vienna Standard Mean Ocean Water (VSMOW). Relationships between daily discharge and streamwater isotopes were assed using Kolmogorov-Smirnov and Spearman's rank correlation coefficients.

Local Meteoric water lines (LMWL) were derived using daily precipitation isotope values from Steglitz (February 2019 – September 2023) and Hasenfelde (July 2018- July 2023) for the urban and rural catchments, respectively, by weighting respective precipitation inputs. To assess evaporation effects on stream water isotopic composition we also calculated the line-conditioned excess (lc-exc), which defines residuals from the LMWL (Landwehr & Coplen, 2006). For each catchment, lc-exc was estimated as:

Urban:

$$lcexc = \delta^2H - 7.8 * \delta^{18}O - 7.1 \ (R^2 = 0.98, p < 0.001) \tag{1}$$

Rural:





$$lcexc = \delta^2H - 7.68 * \delta^{18}O - 7.68 \;\; (R^2 = 0.98, p < 0.001) \hspace{4cm} (2)$$

**3.3 Water ages and mean transit time estimations**

To assess the fraction of stream water that fell as recent precipitation as a metric of the age of stream water, young water contributions ($F_{yw}$) were estimated using the open-access code of von Freyberg et al. (2018). This is based on an iteratively re-weighted least square fitted sine-wave method, using observed precipitation and stream water isotopes to estimate the fraction of stream water that fell as precipitation within previous 2-3 months as an indicator of catchment function. We compared sine-wave fit amplitudes of daily amounts of weighted precipitation $\delta^2H$ and $\delta^{18}O$ to stream water isotopes in both streams. For
simplicity only results for $\delta^{18}O$ are shown in subsequent plots.

For inter-comparison between the rural and urban stream functioning, mean transit times (MMT) were estimated as another metric of hydrologic response. We used daily amount-weighted precipitation isotope data from SUEO (urban) and AWS Hasenfelde (rural), and daily stream water isotopes and applied two different lumped convolution integral models - the three parameter Two Parallel Linear Reservoir Model (TPLR) (Weiler et al., 2003) and the two parameter Gamma Model
(Hrachowitz et al., 2010), to estimate transit time distributions (TTDs) (see Supplementary Table S3 for details). The TPLR model accounts for fast ($\tau_f$) and slow flow reservoirs ($\tau_s$), approximating younger and older water contributions. The reservoirs are partitioned by the $\varphi$ parameter, ranging from 0-1, and which separates the rapid and slow responding flows from surface and subsurface sources. In the urban area we based $\varphi$ on the percentage of impervious area, driving fast urban drainages to the stream. In the rural catchment we used non-irrigated arable land area, as rapid runoff is more likely contributed due to more
compacted soils and agricultural drainage networks.

The Gamma Model is defined by the shape parameter $\alpha$ (-) and the scale parameter $\beta$ (days), with the MTT calculated as the product of the two. The parameter ranges for the TPLR ($\tau_f$, $\tau_s$) and Gamma model ($\alpha$,$\beta$) were sampled from pre-defined parameter ranges using Monte Carlo realizations, in order to find the best fit estimates. To avoid influence of evaporative fractionation on model estimates, a lc-excess filter was applied to stream water isotopes, whereby samples with strong
evaporative fractionation were excluded from calibration. We used different lc-excess filters for urban and rural streams (rural: lc-excess < -2.5‰ (i.e. more enriched); urban: lc-excess < -4‰ (more depleted)), as greater evaporative fractionation effects were observed in the rural stream. Model fits were assessed using Nash Sutcliffe efficiency (NSE) (Nash & Sutcliffe, 1970), root-mean-square-error (RMSE), and coefficient of determination ($R^2$). Due to the limitations of the stable water isotopes to detect transit times longer than 5 years (Stewart et al., 2010), the scale parameter $\beta$ of the gamma model and the $\tau_s$ parameter
in the TPLR model, were limited to 1825 days.



## 4 Results

### 4.1 Differences in rainfall-runoff characteristics

The climatic and hydrological parameters for both catchments contrasted markedly and were summarized for each hydrological year (Table 1). The sampling period remained characterized by relatively dry conditions, following the intensive
drought year of 2018. with subsequent years having below average precipitation. The rural catchment experienced severe drought conditions over 2018-19, with an annual precipitation of 389 mm. Similarly, 2019-20 and 2021-22 were characterized by below average precipitation totals, ~20-30% below the long-term average. Several intense convective summer events and late winter rains in 2020-21 and 2022-23 resulted in precipitation totals closer to the long-term average.

Annual streamflow was sensitive to low annual precipiatation, especially following the 2018-2020 dry period where
average daily flows were ~ 0.01 $l\,s^{-1}\,km^2$. Overall, the rural stream was characterized by clear seasonal intermittency with the onset of the flowing and fully connected phase strongly tied the onset of autumn/winter rainfall, and associated rising groundwater levels. When flowing, average discharge was relatively low (0.15 $l\,s^{-1}\,km^2$), with baseflow comprising 50-70% (BFI: 0.5 – 0.7) of daily discharge. High-flows reached up to 0.5 $l\,s^{-1}\,km^2$ in spring 2022, consistent with elevated groundwater levels during the same winter/spring period (Fig. 2b). Annual runoff was characteristically low, with annual runoff coefficients
between 0.1 to 0.26, owing to the high evapotranspiration. Groundwater levels showed strong seasonality, with relatively shallow depths of around 2.4 mbgl (± 0.3m), with highest levels in April and lowest in autumn. A slight trend in recovery has been noted in 2022 and 2023, following a wetter year.

In the urban catchment, precipitation was generally below the long-term annual mean ~10-20%. Notably, heavy convectional rainstorms during the summers of 2021 and 2023 resulted in above average annual totals (> 600mm/yr).
Characteristic of intensely managed urban streams, a higher specific discharge was observed compared to the rural stream, ranging between 0.2-0.45 $l\,s^{-1}\,km^2$, with peak flows reaching 3.4 $l\,s^{-1}\,km^2$ (Fig. 2d). Consistently higher baseflows were also evident (~ 0.29 $l\,s^{-1}\,km^2$) contributing up to 80% of annual discharge. Groundwater in the unconfined urban aquifer was relatively shallow, between 2-3 mbgl, with only small seasonal differences, but a notable overall decline of summer levels in recent years due to the lack of significant winter recharge. Annual runoff was distinctly higher, generally exceeding the annual
precipitation input, thus reflecting the water subsidy from waste water effluent. This represents inter-basin water transfers (from the Spree and Havel) as abstractions from bank filtration provide Berlin's water supply. This was also reflected in the high annual runoff coefficients (0.8-1.5).


**Table 1: Summarized annual discharge statistics for the rural and urban stream per water year; total annual precipitation (mm),**
**catchment normalized mean ($Q_{mean}$), maximum ($Q_{max}$) and minimum discharge ($Q_{min}$ ) (all in ls$^{-1}$km$^2$) and 95$^{th}$ and 5$^{th}$ percentiles ($Q_{95}$, $Q_5$), as well as annual runoff coefficient (Q/P), annual baseflow index (BFI) and total annual runoff (mm).**

| Water Year | Annual P (mm/wy) | $Q_{max}$ (ls$^{-1}$km$^{-2}$) | $Q_{min}$ (ls$^{-1}$km$^{-2}$) | $Q_{mean}$ (ls$^{-1}$km$^{-2}$) | $Q_5$ (ls$^{-1}$km$^{-2}$) | $Q_{95}$ (ls$^{-1}$km$^{-2}$) | $Q_{Ro}$ (mm/wy) | *BFI* (-) | *Q/P* (-) |
|---|---|---|---|---|---|---|---|---|---|
| **Rural** | | | | | | | | | |
| 2019 | 388.6 | 0.16 | 0 | 0.01 | 0.06 | <0.01 | 44.6 | 0.54 | 0.11 |
| 2020 | 494.4 | 0.23 | 0 | 0.02 | 0.13 | <0.01 | 52.5 | 0.72 | 0.11 |
| 2021 | 534.8 | 0.39 | 0 | 0.04 | 0.18 | <0.01 | 133.7 | 0.58 | 0.25 |
| 2022 | 434.1 | 0.50 | 0 | 0.04 | 0.18 | <0.01 | 113.1 | 0.64 | 0.26 |
| 2023 | 535.2 | 0.33 | 0 | 0.05 | 0.22 | <0.01 | 115.3 | 0.56 | 0.22 |
| **Urban** | | | | | | | | | |
| 2019 | 546.4 | 1.1 | 0.14 | 0.45 | 0.70 | 0.2 | 819.4 | 0.76 | 1.49 |
| 2020 | 527 | 1.2 | 0.01 | 0.27 | 0.54 | 0.07 | 670.5 | 0.72 | 1.27 |
| 2021 | 600.5 | 2.3 | 0.04 | 0.45 | 0.67 | 0.19 | 872.8 | 0.78 | 1.45 |
| 2022 | 461.4 | 2.1 | 0.06 | 0.35 | 0.65 | 0.14 | 447.4 | 0.86 | 0.96 |
| 2023 | 624.0 | 3.4 | 0.01 | 0.33 | 0.62 | 0.165 | 523.0 | 0.82 | 0.84 |

## 4.2 Isotope dynamics of precipitation and streamflow

The isotopic composition of rainfall in both catchments was highly seasonal (rural: mean $\delta^{18}$O = -6.03‰, SD=3.27‰ and mean $\delta^2$H = -42.12‰, SD=21.59‰; urban: mean $\delta^{18}$O = -6.09‰, SD=3.15‰ and mean $\delta^2$H = -42.11‰, SD=21.97‰), with
rainfall depleted in heavy isotopes in winter and more enriched in summer (Fig.2a); with occasionally highly enriched signatures following larger summer convective storms. The differences in catchment characteristics and urban influence were reflected in the daily isotopic signatures of the streams (Fig. 2, 3). In the rural stream, distinct effects of evaporative fractionation were evident in early summer samples from ponded water in riparian areas and wetlands, as well as near beaver dams after intense storm events (Fig. 3a). Most notable were the seasonal differences in early streamflow samples from
December and January (mean $\delta^{18}$O = -8.36‰), which was close to groundwater (mean $\delta^{18}$O = -8.7‰, SD = 0.18‰; mean $\delta^2$H = -58.9‰, SD=0.9‰), while later in the season the influence of recent precipitation was more pronounced (mean $\delta^{18}$O = -7.2‰) (also see Supplementary Figure S1). Lc-excess in the rural stream frequently showed negative values and a strong

seasonality (0.8 to -11‰)., indicating greater seasonal fractionation from late spring – early summer during and prior to streamflow cessation.



**Figure 2: Timeseries of daily precipitation and precipitation δ¹⁸O isotopes in the a) rural and c) urban stream as well as timeseries of 15-min discharge and daily streamflow δ¹⁸O isotopes in the b) rural and d) urban stream. Baseflow is indicated in red as part of the discharge timeseries. Groundwater levels are indicated as meters below ground level (mbgl) in grey. Weekly isotopes in the urban stream (orange) are indicated for 2023 (d). Grey shading indicates selected storm events, with winter events highlighted in red and summer convective storms highlighted in green.**

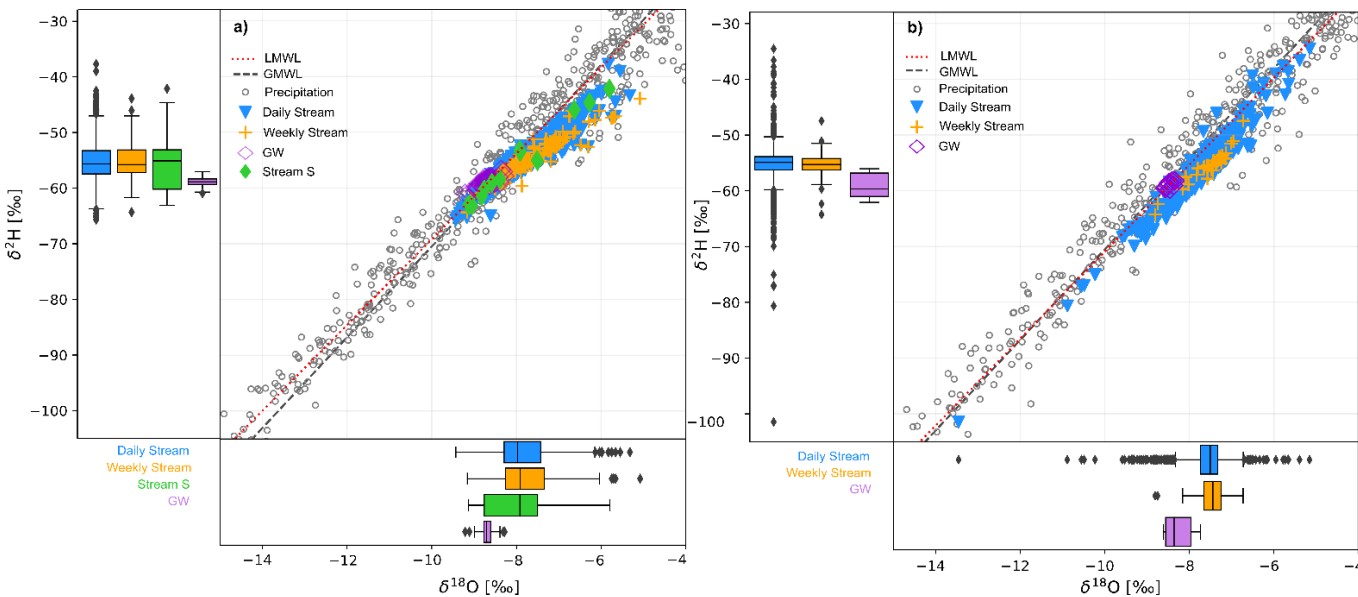

**Figure 3: Dual isotope (center) and boxplots (left, bottom) showing the isotopic composition of daily precipitation (grey), daily streamflow (blue, weekly streamflow (orange) and groundwater isotopes (purple) in the a) rural and b) the urban catchment. Additional isotope samples (Stream S; green) from isolated pools in the stream are shown for the rural catchment. The Global Meteoric Water Line (GMWL, black) and amount-weighted local meteoric water lines (LMWL, red) from precipitation signals are shown for reference.**

The urban stream exhibited more variability in stream water isotope composition, with the majority of the samples plotting below the GMWL (Fig. 3b). The streamflow signature appeared less damped than in the rural stream, with occasional pronounced responses urban storm drains, which is as important as the limited temporal variability (Fig. 2b). Occasionally more enriched signals in the stream can be seen in summer (mean $\delta^{18}O$ = -7.4‰), likely attributable to an increased inflow of enriched effluent. The most depleted signals were observed after intense convective storms (i.e. July 2021, $\delta^{18}O$ = -13.46‰).

The variable inflow of differently fractionated water was visible in the lc-excess signal, which showed clear variability (-7.9 to 4.3 ‰), with lower lc-excess values estimated in winter, reflecting the inflow of more fractionated sources (e.g. waste water, lake water), and only occasional positive lc-excess values following rain events (see Supplementary Figure S3. Groundwater from the Panketal aquifer generally also had a damped and stable signature with low variability (mean $\delta^{18}O$ = -8.46‰, SD=0.07‰; mean $\delta^{2}H$ = -58.98‰, SD=0.53‰).

**4.3 Differences in seasonal flow regime**

       Annual flow duration curves (FDCs) showed different inter-catchment sensitivities of flow regimes and high flow conditions (Fig. 4a). In the rural stream, the higher slope of flow duration curves highlighted its intermittent nature and low baseflow component, with most years showing zero flow periods for an average of 60% of the year. The effects of the drought are reflected in the flat curves of 2018-19 and 2019-20, where low or no flow conditions ($Q_{95}$) were exceeded almost 70% of



the year. In contrast, high flow conditions ($Q_5$) were greatest during 202-23 following a wet spring, and flows continued for almost 65% of the time. The mean flashiness index for the rural stream was relatively low at 0.07.

**Figure 4: a) Annual flow duration curves per water year using daily streamflow data for the rural (dotted lines) and urban (solid lines) streams. The stream in the rural catchment shows a characteristic intermittency with much higher likelihood of drying, while the urban catchments clearly perennial with only ephemeral high flows. b) and c) Relationships between daily stream water $\delta^{18}O$ and specific discharge for the rural and urban stream respectively. Horizontal lines indicated mean measured groundwater (red) and mean precipitation (blue) end member $\delta^{18}O$.**

In the urban stream, flow duration curves were indicative of the more perennial nature with high variability and stronger baseflow component, and more marked responses to precipitation events as evidenced also in the runoff coefficients. However, the flashiness index was also relatively low (0.14). Similar to the rural stream, the effects of the 2018-19 drought were clearly

evident, with specific discharges of low flows (<$Q_{95}$) being highest in WY 2019, corresponding to a relatively dry winter and considerable baseflow augmentation through increased effluent contributions. Similarly, specific discharge was higher during low flows in WY 2021, corresponding to the observed higher annual baseflow ($Q_B \sim 0.35$ ls$^{-1}$km$^2$). The flashy streamflow responses to summer convectional events during WY 2022 and 2023 were visible in the partially steeper slope of the medium





flow segment (0.2-0.7 flow exceedance probabilities), while the flatter mid-segment in WY 2020 hints at a more sustained

groundwater flow contributions, corresponding to the increased groundwater levels observed in the Panketal Aquifer (AQ 1.2).

Relationships between daily isotopic variations (indexed by $\delta^{18}O$) and specific discharge provides further evidence of time-variant source contributions to streamflow in each catchment (Fig. 4b, c) In the rural stream, $\delta^{18}O$ showed a positive correlation with discharge (r= 0.41, p <0.01), despite overall modest variations in daily $\delta^{18}O$ (SD= 0.6‰). A marked seasonality could be seen between winter flows, which are generally more depleted, and spring flows where, as streamflow

increases, stream water isotopes became slightly more enriched and relatively stable, indicating time-variant contributions from mixed sources (e.g. precipitation and groundwater). Towards summer, as streamflow starts to decline, the remaining stream water was increasingly subject to evaporative fractionation effects particularly as groundwater levels fall and the channel network becomes disconnected leaving isolated open water surfaces in the streambed (Fig. 4b). In the urban stream, $\delta^{18}O$ and discharge were negatively correlated (r= -0.25, p<0.01) with a clear seasonal distinction between high flows in winter

and low flows in summer. Furthermore, negative $\delta^{18}O$ anomalies could be observed all year round, while positive anomalies were summer based.

### 4.4 Stream responses to individual storm events

The climatic differences between sites became apparent in some of the rain events, resulting in contrasting storm totals and streamflow responses. Table S2 in the Supplementary Material summarizes the characteristics of selected storm events.

Winter (October-May) and summer (June-September) storm events were characterized by different rainfall amounts and discharge responses, with the most intense storms ($P_{int,5}$) predominantly occurring in summer. Most notably, during summer 2019 a large convectional event over Berlin produced almost 46 mm/hr and an urban peak discharge of 5.3 ls$^{-1}$km$^2$, while in rural Brandenburg no rain was recorded during the same period. The extremely sandy soil of the rural catchment and large soil moisture deficits resulted in a limited response of streamflow to summer storm events, despite the size and intensity of some

of the events (i.e. June 2021). These produced only minimal transient flow, or very small streamflow peaks (i.e. summer 2020) from saturated areas fringing the channel bed or the wetter riparian areas upstream, and receding quickly to isolated pools of water, leading to highly fractionated signatures in summer stream water isotopes (Fig. 2b). Furthermore, runoff coefficients were negligible during summer. Nevertheless, the wet period of spring 2023 produced significant rainfall over multiple days and elicited higher discharge responses in April and May (up to 0.3 ls$^{-1}$ km$^2$), resulting in increased runoff coefficients (0.2)

and recharge responses, as well as streamflow until early summer.

More intense rain events were observed in the urban catchment (49.8 mmhr$^1$) compared to the rural catchment (19.6 mmhr$^{-1}$), with urban precipitation totals of up to 55mm in less than 4 hours observed in summer 2021. In the urban stream, the larger, more intense summer convectional events resulted in significant discharge peaks, whereas negligible streamflow responses were detected in the rural stream. A strong positive correlation was found between storm discharge ($Q_S$) and total event

precipitation ($R^2$= 0.54, p<0.0001), as well as mean precipitation intensity ($P_{int}$) ($R^2$= 0.44, p<0.0001) in the urban stream. In



contrast, in the rural site event discharge was less strongly correlated to event precipitation ($R^2 = 0.18$, p<0.001). Due to the nature of the urban catchment functioning, distinct and large step changes with differences of up to 10 ls$^{-1}$km$^2$ were observed following the most intense events. Hydrograph separation showed that the large streamflow peaks after events typically immediately subsided (within < 1hr). Conversely during low flow and drought periods, step changes by up to 1 m$^3$/s (from 0.5 to 1.5 m$^3$/s) could be observed (i.e. April 2020) due to baseflow management. During peak flows, stream isotope anomalies were in the same direction as event precipitation, as seen in severely depleted signatures following extreme convective events, with the rainfall signal persisting over several days in the case of the most intense events (Fig. 2d). Event runoff coefficients varied with the size and intensity of the events, with the highest coefficient observed in during the wet period in autumn of 2019 and 2020, during which time flows were frequently diverted out of the catchment to reduce flood risk.


**4.5 Inferring stream water age and transit times from isotope data**

Estimates of water ages and average young water fractions ($F_{yw}$) (in this case % of water younger than 2-3 months) are shown in Figure 5. In the rural catchment, average young water fraction was ~ 15%, with a statistically significant fit (p<2.2e-16, $R^2_{adj}$=0.41, RMSE=0.25) (Fig. 5a). For water years individually, the model suggests some interannual variability between wetter and drier years, with young water contributions ranging from ~ 10 up to ~35% (Table 2). Mean annual young water fractions were negatively correlated to annual precipitation (r= 0.67, p<0.01) and annual discharge (r= 0.61, p<0.01). Most notably for WY 2019, despite low runoff and overall low precipitation, $F_{yw}$ was surprisingly high at around 37%, consistent with a low BFI (~0.5). At the same time for WY 2023, $F_{yw}$ was also estimated to be around 30%, which would be more consistent the increased contributions of recent precipitation to streamflow, during the wet spring and intense summer convective events. Conversely in WY 2021, which was a similarly wet year in terms of total annual P (~540 mm/wy), $F_{yw}$ was estimated at only 10%, despite several large storm events (e.g. June 2021).



**Table 2: Young water fractions over the entire study period and per water year for the rural and urban stream obtained from sine-wave fitting, including coefficient of determination ($R^2$), p-value and residual standard errors over the entire study period.**

| WY | $\delta^{18}O$ Rural | | | | $\delta^{18}O$ Urban | | | |
|---|---|---|---|---|---|---|---|---|
| | $F_{yw}$ | $R^2_{adj}$ | p-value | RSE | $F_{yw}$ | $R^2_{adj}$ | p-value | RSE |
| **2019** | 0.37 | 0.83 | <2.2e-16 | 0.14 | - | - | - | - |
| **2020** | 0.22 | 0.70 | <2.2e-16 | 0.25 | 0.12 | 0.35 | 3.82e-13 | 0.22 |
| **2021** | 0.09 | 0.58 | <2.2e-16 | 0.17 | 0.05 | 0.05 | 0.0002 | 0.33 |
| **2022** | 0.24 | 0.62 | <2.2e-16 | 0.15 | 0.05 | 0.08 | 3.67e-07 | 0.23 |
| **2023** | 0.28 | 0.37 | <2.2e-16 | 0.23 | 0.15 | 0.21 | 0.0002 | 0.24 |

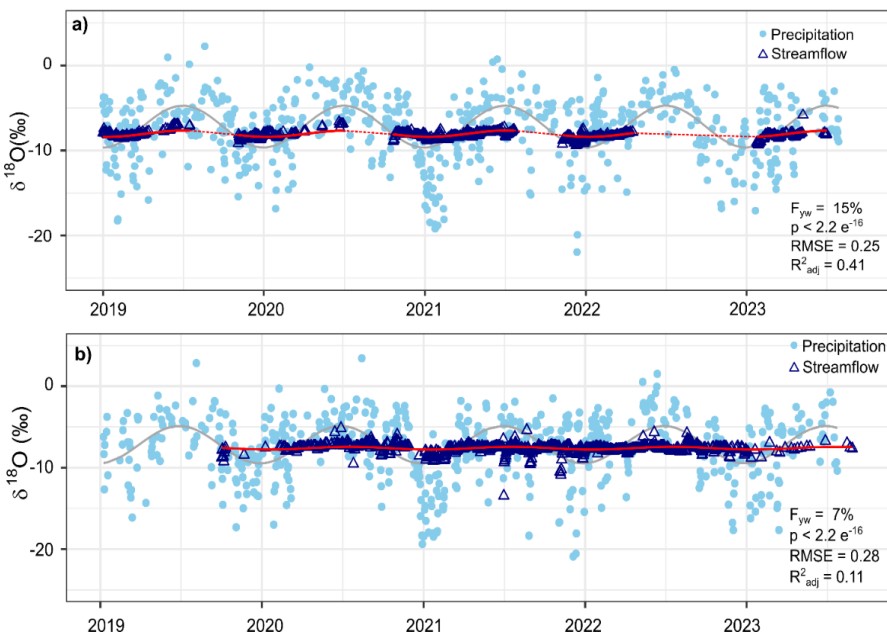

**Figure 5: Hydrologic and isotopic seasonality of precipitation (light blue) and streamflow (dark blue) for the a) rural and b) urban stream, respectively. Sinusoidal cycles (red) were fitted to daily stream isotope data using IRLS (after von Freyberg et al., 2018) for estimates of recent water contributions (<2-3 months). Periods of now flow in the rural catchment are noted with dotted lines in the sinusoidal fitting.**

In the urban stream, despite the significant amounts of urban storm drainage, young water estimations were relatively low, averaging around 7% (p<2.2e-16, $R^2_{adj}$=0.11, RMSE=0.28) (Fig. 5b). Inter-annual variability was generally limited, with estimates varying only between 5% to 15%, an indication of the time-variant contributions of younger water that includes precipitation as well as recent waste water discharge. Mean annual young water fractions were positively correlated with annual precipitation totals (r=0.47, p<0.01) and negatively correlated with annual discharge (r =0.49, p<0.01). Similar to the rural stream, the rain-intensive period of early 2023 and subsequent intense summer convective events resulted in an increased young water contribution of up to 15%. $F_{yw}$ was lowest in water years 2021 and 2022, with estimates of around 5%, consistent with higher BFI values (~ 0.8).

The TTDs from the TPLR and gamma models were fitted successfully to both streams (Fig. 6, see Supplementary Table S4 for further model results). The TPLR model gave slightly better fits for modelled $\delta^{18}$O values than the Gamma Model in terms of NS statistics and $R^2$ (Table 4). Estimated MTTs were distinctly higher for the TPLR, ranging between 3 to 4 years for the rural and urban stream, respectively, while MTTs with the gamma model were only estimated to be less than 2 years for both streams. The median fast and slow transit times ($\tau_f$/$\tau_s$) were 22/1262 days for the rural stream and 5/1311 days for the urban stream. Despite inherent limitations in the detectability of stream water $\delta^{18}$O signatures older than 5 years, the TPLR model was useful in capturing the rapidly responding flowpaths through the fast component, which drives streamflow flashiness in both streams. The fast flow contributions were also similar to the range of $F_{yw}$ estimates (rural: $\tau_f$ ~ 22d, $F_{yw}$ ~ 15%; urban: $\tau_f$ ~ 5d, $F_{yw}$ ~ 7%). Nevertheless, especially in the rural stream, both models fail to capture the streamflow

signatures towards summer, especially during the 2019 period, but appear to work well for the wet periods of 2023. Furthermore, in early 2021, the influence of the more depleted winter precipitation on streamflow was overestimated. The more damped isotopic dynamics of the urban stream were generally well captured in both models, with only the most short-term responses to the severely depleted high intensity convectional events (i.e. summer 2021) and the occasional inflow of

more enriched sources (e.g. effluent), not fully captured.

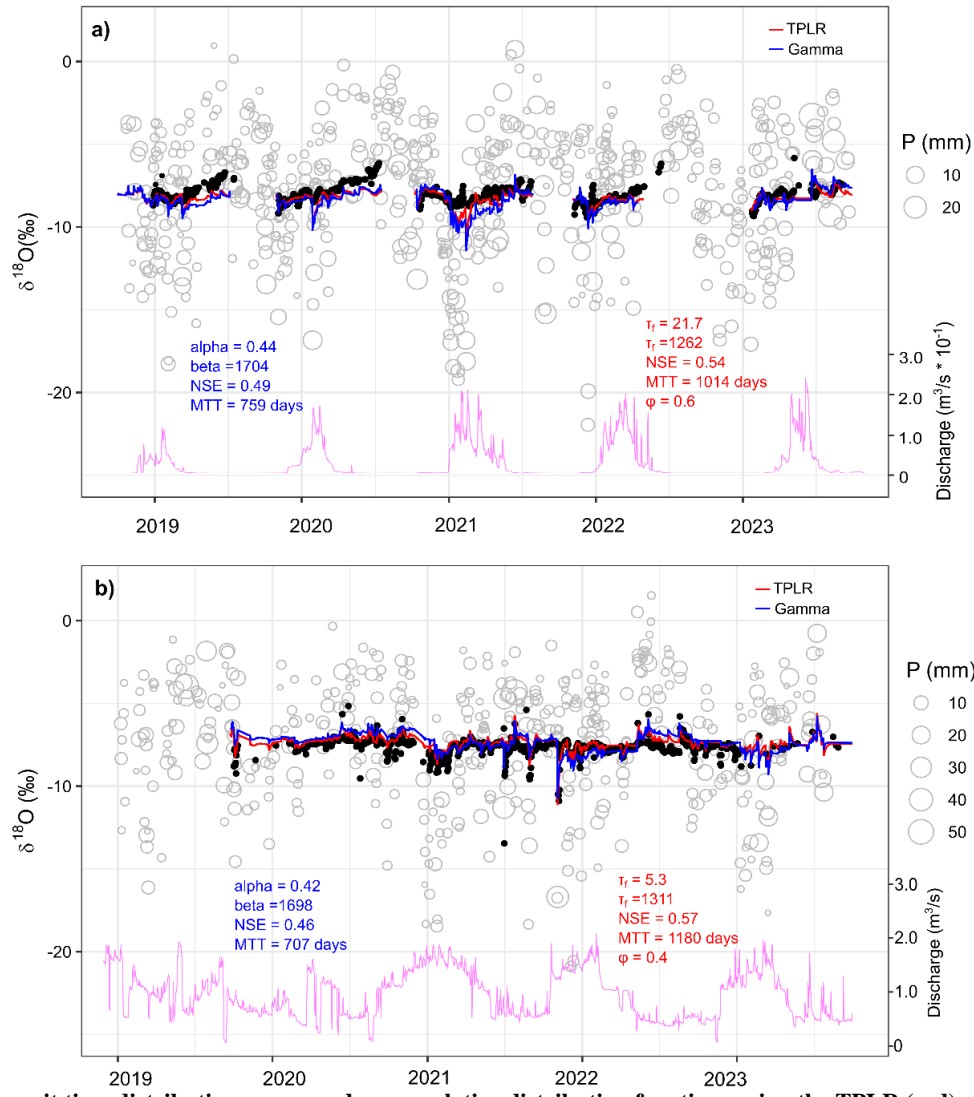

**Figure 6: Fitted transit time distributions expressed as cumulative distribution functions using the TPLR (red) and Gamma model (blue) for the a) rural and b) urban catchment. Precipitation amount (in mm) is denoted by the size of the open circles, while isotopic values of precipitation inputs correspond with the y-axis (‰). Periods of no flow in the urban catchment have been indicated with a gap in the simulations. Daily discharge is shown in pink (m³/s).**





## 5 Discussion

### 5.1 Seasonal and event flow responses

Recent severe drought hazards across central Europe have highlighted the vulnerability of streams to the compound effects of drought progression on frequency of intermittency and longevity of zero-flow periods (Creutzfeldt et al., 2021; Sarremejane et al., 2022; Tramblay et al., 2021). In Berlin – Brandenburg, the 2018 drought and sustained negative rainfall anomalies, exacerbated a decade-long trend of declining groundwater levels, causing increasing dis-connectivity between groundwater and surface water and raising concerns regarding future water availability (Creutzfeldt et al., 2021; DWD, 2019). Propagation of precipitation deficits into groundwater droughts is particularly detrimental in areas where groundwater is the main contributor to baseflow, as in the case of the rural stream (up to 70%). Consequently, flow regimes in similar drought-sensitive lowland areas will likely become more vulnerable to hydroclimate hazards and landuse change impacts on terrestrial hydrological processes, which drive soil moisture availability and groundwater recharge (Luo et al., 2024; Paredes del Puerto et al., 2024; Van Loon & Laaha, 2015; Wunsch et al., 2022).

Compared to the long-term mean of discharge in the rural catchment (0.11 m$^3$/s; 2001-2017, Ying et al., 2024), daily discharge has declined by up to 60% following the 2018 drought. Due to the limited replenishment of moisture during winter in 2018-19, the observed low discharge and longer no-flow periods defined the 208-19 hydrological year. Even in subsequent years, the lack of summer streamflow response to even the largest precipitation events (e.g. June 2021), coincided with observed soil moisture deficits as well as limited groundwater recharge in the ongoing drought. This limited the seasonal variation in surface water connectivity and streamflow generation (Landgraf et al., 2022; Ying et al., 2024), and also affected nutrient transport and chemodynamic behaviour, as low flows decrease transport capacity and dilution effects (Wu et al., 2021).

Sensitivity of the blue water fluxes of groundwater and streamflow generation to droughts has been noted ubiquitously across various lowland catchment environments, where the influence of landuse on water partitioning and hydrologic connectivity define catchment functioning (Natkhin et al., 2012; Paredes del Puerto et al., 2024; Smith et al., 2020; Wegehenkel, 2009). Using daily isotopes evidenced the close links between seasonal groundwater and the relative elasticity in streamflow responses to drought years and the timing of precipitation (for more detail see Supplementary Figure S1,S2). While the overall variability of the isotopic composition of streamflow was low, the role of seasonal precipitation in maintaining streamflow connectivity and the importance of groundwater for winter streamflow generation, clearly define the seasonal isotopic signal. Nevertheless, increasing ET and projected precipitation decreases may compromise the catchments' ability to maintain green and blue water fluxes throughout the year in areas with lower water retention. Especially in agricultural catchments, the dominance of green water fluxes from water intensive crops such as maize and other cereals, can propagate inter-year drought memory effects from reduced blue water fluxes to groundwater recharge and streamflow generation (Ihinegbu & Ogunwumi, 2022; Orth & Seneviratne, 2013), creating uncertainty over the resilience of agricultural production and landuse (Beillouin et al., 2020; McNamara et al., 2024)



Ongoing precipitation deficits after the unusually dry winter in 2019 also reduced urban stream baseflows throughout the metropolitan area of Berlin (Kuhlemann et al., 2020; Marx et al., 2023). While baseflows in the urban stream were only slightly lower during drier years, overall discharge was governed by increased discharge of WWTP effluents, regularly accounting for

up to 90% of total discharge (Marx et al., 202), underscoring the crucial role of effluents in sustaining baseflows during extreme drought periods (Luthy et al., 2015). While in the upper catchment, the stream exhibits naturally intermittent behaviour, the absence of clear seasonality in the flow regime, due to baseflow regulation and runoff following  storm events, highlights the persistent influence of urban water management on rapid flowpaths, inter-basin water transfers and streamflow generation. (Bhaskar et al., 2016). As such, the mechanisms affecting urban groundwater recharge and baseflow are not only shaped by

urban infrastructure and development, but are increasingly driven by natural hazards such as floods and droughts, which challenge not only water availability resilience but managers' ability to tailor resource management approaches to local climate conditions (Hale et al., 2016).

The unexpectedly low flashiness index in the urban stream did not directly reflect the peak flow responses observed in individual stream hydrographs, as the stormflow peaks generally dissipated within less than 1 hour due to flood mitigation

actions. The low flashiness is likely explained by the relatively constant baseflow but also the broader interaction of the stream network with wetland storage, floodplains and inter-basin transfers out of the catchment, which shape the downstream propagation of the flood peak through the urban catchment (Johnson et al., 2022; Oswald et al., 2023). In future, this infrastructure gives managers flexibility in times of increased drought and flood hazards and can strengthen urban resilience to natural hazards. Additionally, resilience can be built by preserving and enhancing wetland areas in peri-urban and rural

areas, increasing infiltration and storage of excess surface water through sustainable urban drainage systems and optimizing urban water management within the built environment through nature-based solutions (Davis & Naumann, 2017; Green et al., 2021). The successive disconnection of urban drainage and green space requirements for new urban developments has already been set as a key measure for future urban development, opening up avenues for blue-green infrastructure and nature-based solutions to create synergies between flood management and urban cooling (SenUVK, 2022)

**5.2 Transit times and water ages**

Multi-year daily isotope data provided a unique dataset to identify runoff sources and quantify transit times in the two catchments. Although estimates of young water fractions, MTTs and TTDs only provide a first approximation of complex age and transit time distributions, due to limitations of the $F_{yw}$ approach (Kirchner, 2016; Seeger & Weiler, 2014) and the inability of lumped convolution models to account for evaporative fractionation (McGuire & McDonnell, 2006), we gained insights

into the differential catchment functioning between the two streams in response to hydroclimate variability and catchment characteristics. Further, estimates of water ages in both streams highlighted the contrasting storage-discharge relationships between a highly urbanized and non-urbanized groundwater dominated stream environment.

Previous estimates from only <2- years of daily data, indicated high levels of uncertainty in MTT and $F_{yw}$ estimates in the rural catchment (Kleine et al., 2021). The longer, continuous timeseries of daily stream and precipitation isotopes used in this





study and the systematic filtering of fractionated signals of early summer streamflow not only revealed inter-annual differences in young water contributions related to annual precipitation (negative correlation), but also improved young water fraction estimates for different years with lower values between 0.1 and 0.37 (mean 0.15) compared to previous estimates (mean 0.37, Kleine et al., 2021). These values are more consistent with estimates from other lowland catchments in central Germany (Lutz et al., 2018) and only slightly below the median young water fraction of 26% for European catchments (Jasechko et al., 2016).

The larger $F_{yw}$ in the rural stream (vs urban) may be attributed to the different landuse and soil characteristics and the relatively small catchment size, whereby the predominantly agricultural landuse (drainage and compaction) facilitates rapid runoff responses via the artificial drainage pipes and channels control the release of young water contributions to the stream from relatively small areas in the catchment (Lutz et al., 2018; Von Freyberg et al., 2018). While, in general, young water fractions tend to be higher when catchments are wetter and discharge is higher (Von Freyberg et al., 2018), the negative relationship

between annual precipitation and young water fractions may indicate more complex hydrological interactions that control routing near the land surface during drought periods. In wet years such as 2023, $F_{yw}$ was higher as a result of active drainage and compacted agricultural soils in the upstream part of the catchment created localized surface/near-surface flowpaths and a rapid response of event water during wet periods. However, during the drought, lack of storage to mix with intense rainfall events can quickly route rainfall signals towards the stream at a time when groundwater inputs are low due to the due to the

lack of recharge. hydromorphic conditions of severely dried out soils, thus leading to higher $F_y$ during the driest year. However, it could also be that these contradictions are partly artefacts of the method, as the data collection period was still quite short due to the stream's intermittency increasing the uncertainties associated with the non-stationarity of TTDs over different time scales in response to hydroclimatic conditions (Hrachowitz et al., 2010). As a result, particularly during the drought this may produce a less robust seasonal cycle coefficient for each individual hydrological period, as evidenced by the overall higher

uncertainty attached to $F_{yw}$ estimates in drier years.

The relatively low young water fractions of ~ 7 % in the urban stream, despite significant urban runoff, highlight the complicating issue of constraining water ages in urban streams, where an overwhelming dominance of mixed urban sources weakens the influence and differentiation of recent precipitation (Bonneau et al., 2018; Soulsby et al., 2014). This is a particular impediment in closed urban water management systems, such as Berlin, where there is overlap in the isotopic composition of

surface waters, groundwater and wastewater contributions (Massman et al., 2008). However, the positive correlation with annual precipitation is broadly consistent with the higher impact of precipitation events during the winter and spring season, and greater urban drainage and fast flowpaths that are activated in response to intense summer precipitation, as seen in the highest $F_{yw}$ (~15% ) in 2023 (Marx et al., 2021; Von Freyberg et al., 2018).

Regarding TTDs and estimated MTTs, both models (gamma and TPLR) provided reasonable approximations of the

isotope responses with similar uncertainty, but better fits with the TPLR. The estimated MTTs were surprisingly similar for both streams, although diverging between methods. For the rural stream, our estimates exceed previous estimates by Kleine et al. (2021), which were deemed unrealistically low. Between models, our estimates range between 2-4 years (730 days – 1622 days), with the higher estimate from the TPLR model more likely, based on recent estimates of local groundwater with tritium





ages of around 5 years (Ying et al., 2024) and stream water ages of ~7 years derived from tracer aided models (Smith et al.,
2021). Furthermore, similar to previously noted limitations in MTT estimations in lowland areas (Tetzlaff et al., 2011), the
pervasive influence of older groundwater in the rural stream likely weakens the influence of precipitation intensity on MTTs,
as evidence by the occasional overprediction of isotopic response in the stream during early flow periods in winter (e.g. early
2021) (Fig. 6a). The better performance of the TPLR compared to the Gamma model underlines the particular suitability of
the two-reservoir model in an urban environment, where runoff generation is a more binary distribution of processes (Soulsby
et al., 2014). The estimated MTTs of ~ 4 years in the urban stream are slightly longer than estimated in previous work
(Kuhlemann et al., 2022).

## 5.3 Wider Implications

With the projected changes for Europe suggesting a greater seasonal divergence of lower and higher precipitation amounts
in summer and winter respectively, the seasonal synchronicity of groundwater-baseflow responses we found in our study may
widen even further, as the longer time scales for groundwater recovery after extended precipitation deficits usually lags behind
several years despite the return to wetter conditions (Hellwig et al., 2020; Smith et al., 2022). The effects of such hydroclimate
changes may be even more severe in urban areas, where urbanization has directly been linked with the intensification of
extreme rainfall (Singh et al., 2020). Urban "plumbing", a high level of imperviousness and lack of urban green space, can
reduce recharge and groundwater storage to the point where they are unable to buffer natural climate variability (Bonneau et
al., 2018; Marx et al., 2021; Oswald et al., 2023).

The differential impacts of drought and extreme events and landuse on streamflow generation are conceptualized in Figure
7, illustrating the seasonal controls on flow permanence and magnitude of associated hydrological processes in contrasting
environments. During drought periods with reduced rainfall and high ET, severe soil moisture deficits and low recharge
increase the disconnection between surface and groundwater, leading to long-lasting groundwater droughts, reduced
agricultural production and increased streamflow intermittency. Conversely, in the urban environment these effects are
moderated through the supply of waste water effluent or other continuous sources, but which in turn can lead to increased
nutrient concentration and negative impacts on freshwater quality and diversity (Numberger et al., 2022; Warter et al., 2024).
In the context of natural hazards and climate change, more frequent and intense rain events and flooding, can challenge urban
infrastructure and inner-city drainage systems, as greater quantities of water also increase organic pollutant loads and threaten
aquatic habitats and biodiversity (Creutzfeldt et al., 2021; Haase et al., 2023).

Diverging perceptions between stakeholders of the wider implications of water-related risks on the provision of ecosystem
services requires greater understanding of hydrological change and developing systemic approaches that utilize the co-benefits
of nature to mitigate  impacts of emerging climate hazards as well as increasing the resilience of human-environment systems
to water-related hazards (Bush & Doyon, 2019; Green et al., 2021; Raymond et al., 2017). The direct link between landuse
and streamflow, and recharge dynamics found here illustrates the urgency for landuse managers to consider the sustainability





of current and future landuse transformation in drought sensitive areas, like Brandenburg. (Gutzler et al., 2015; Reyer et al., 2012). Moving away from mono-cultures to more integrative landuses that combine trees and shrubs in the agricultural landscape, can play a major role in mitigating water losses and promoting recharge dynamics (Jacobs et al., 2022).

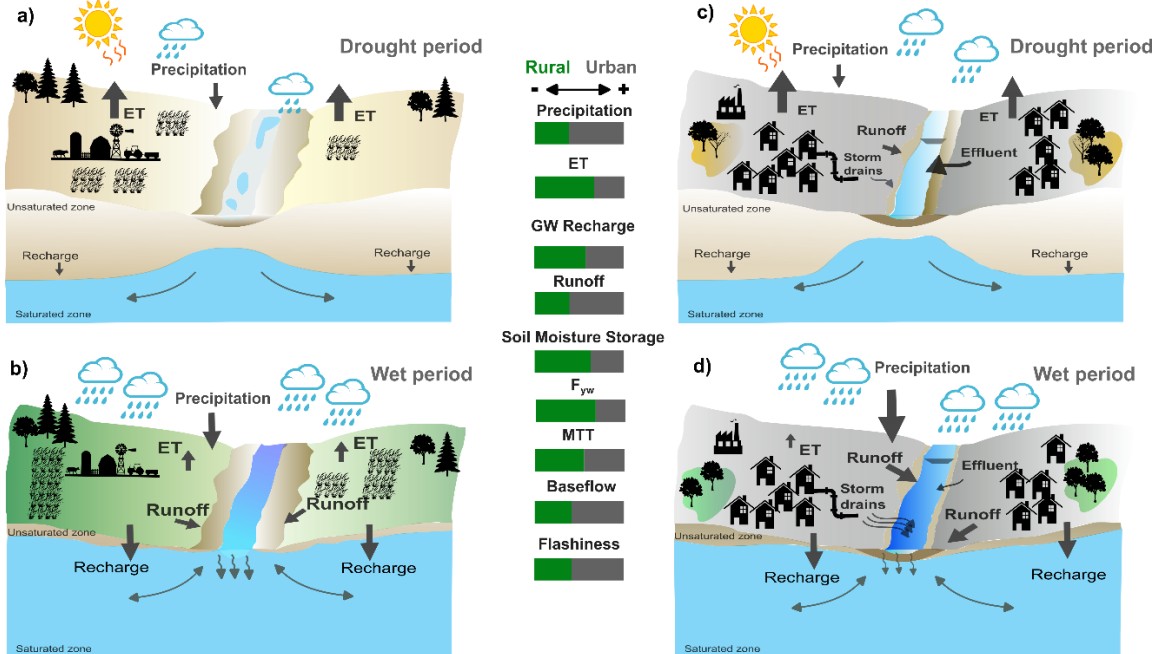

**Figure 7: Conceptual summary of dynamics in hydrological processes and metrics in the rural agricultural and urban catchment during a) and c) drought periods and b) and d) wet periods. Bars in the middle represent general magnitude of fluxes in each catchment in comparison.**

However, while there is a need to reconcile different water use requirements and preserve the hydrological and ecological integrity of anthropogenically influenced streams, this is inherently more difficult with insufficient information on hydrological process dynamics and landuse influences. This issue is particularly prevalent in urban areas, where a comprehensive understanding of basic hydrologic processes across urban water interfaces which encompass engineered management systems and more natural ecohydrological processes in urban green space, is still lacking (Gessner et al., 2014). Emerging questions whether declining baseflows should be seasonally augmented by treated wastewater or inter-basin water transfer are not only a matter of social choice but also need to consider the hydrological, ecological and chemical impacts of increased baseflow contributions on instream habitat, biodiversity targets and water quality (Numberger et al., 2022; Warter et al., 2024). Designing flow regimes to achieve specific ecological and hydrological restoration goals may become the norm in modified and managed rivers where a return to natural, pre-anthropic conditions is no longer feasible (Acreman et al., 2014; Stewardson et al., 2017). As such, identifying thresholds at which important hydrological changes occur, requires a thorough understanding of how water moves through catchment systems.



## 6. Conclusion

570 Inter-catchment comparisons between urban and rural stream systems using multi-year tracer-based assessments are still rare but – as demonstrated here - very insightful and much needed. From the strong responses to drought and an increasing hydroclimatic variability, landuse (i.e. drainage, vegetation, wetland restoration) showed to be important in water partitioning of groundwater-surface water interactions and streamflow generation in these anthropogenically impacted streams  Our results not only highlighted the continued importance and value of high-resolution long-term tracer data to develop a synoptic

575 understanding of the principal hydrologic mechanisms by which flow regimes directly and indirectly respond to climate perturbations, especially in understudied urban environments. They provided immediate evidence of contrasting catchment functioning and streamflow generation in different geographical settings, which will be useful for the identification of future environmental flow assessments in similar urban and lowland catchments where a return to pre-anthropic natural conditions may be no longer attainable. The limitations in the detection of water ages and source contributions in the urban stream

580 highlighted the need for long-term tracer-based assessments of urban hydrological fluxes, using either stable isotopes or other reactive tracers, to better constrain current and future inter- and intra-annual variability and to mitigate the effects of hazards such as floods and droughts. Although the challenges associated with sustained monitoring often limit long-term observations over broader scales, the benefits and value of long-term observations are crucial for hydrologists, ecologists and urban planners and local stakeholders interested in protecting and maintaining ecosystem function and manage future water resources in the

585 most sustainable and integrated way that reduces environmental impact and economic costs.

**Data Availability Statement:**

The data used in this study is available on the Leibniz Freshwater and Environmental Database (FRED) under the following doi: 10.18728/igb-fred-865.0. Data will be made available upon publication. Public discharge data is available from

590 https://wasserportal.berlin.de/start.php. Climate Data from the German Weather Service (DWD) can be downloaded from https://www.dwd.de/DE/leistungen/cdc_portal/cdc_portal.html?nn=17626.

**Acknowledgements:**

Funding for MW and DT was received through the Einstein Research Unit "Climate and Water under Change" from the Einstein Foundation Berlin and Berlin University alliance (grant no. ERU-2020-609) and through BiodivRestore for the

595 Binatur project (BMBF No. 16WL015). CM and CS were also funded by the Einstein Stiftung Berlin, MOSAIC project Grant/Award Number: EVF-2018-425. This study was further funded through the German Research Foundation (DFG) as part of the Research Training Group "Urban Water Interfaces" (UWI; GRK2032/2). We also acknowledge David Dubbert and Franziska Schmidt from the IGB Isotope Lab for help with the isotope analysis, Jonas Freymüller for help with site installations and maintenance of equipment in the DMC and Jan Christopher for help with sampling.


The authors declare that they have no conflict of interest.





**CRediT author contribution statement**

Conceptualization: DT, CS, MMW; Methodology: MMW, CM, DT, CS; Investigation: CM, MMW; Formal Analysis:

MMW; Data Curation: MMW, CM; Writing-Original Draft: MMW; Writing – Review & Editing: DT, CS; Visualization:

MMW; Supervision: DT, CS; Funding acquisition: DT, CS

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
