# Peer review of "Impact of drought hazards on flow regimes in anthropogenically impacted streams: an isotopic perspective on climate stress"

_Natural Hazards and Earth System Sciences, 2024_

## Author Comment (AC1)

**Response to Referee Comment 1:**
Thank you for your constructive comments and suggestions. We believe addressing these comments will strengthen the paper and improve the message and key points we are trying to convey. Below, we respond to the specific comments, point by point and provide clarifications where necessary. We are confident that through this process we can improve the structure and effectiveness of the paper and communicate the results more clearly. Importantly, we are not sure if the reviewer was made aware that this paper was submitted to a Special Issue on the region "Berlin/Brandenburg" with a very specific local focus. Considering the scope of the SI might help to clarify some of the comments made.
Sincerely,
Dr. Maria Magdalena Warter (on behalf of all co-authors)

This paper by Warter et al. deals with the resilience of streams facing droughts. This is a very interesting topic, as this resilience is due to complex processes that are dependent on interacting catchment characteristics (climate, geology, pedology, land use, water management practices,…). The study analyses hydrological and stable water isotopes data from a 5-year data set on 2 contrasted catchments in Germany (sizes, geologies, land uses,…).
The paper clearly has a lot of potential and deals with a large amount of data.
** Thank you for this positive evaluation of our manuscript.

However, I found it very long, wordy, and difficult to read, mainly because it lacks focus and precision in the analysis. Therefore, it is complicated for the reader to appreciate the results and the impact of the paper. My suggestion would be to rework the data to be able to present less «raw» and more to the point results. My main remarks and recommendations are listed below:
**We thank the reviewer for the careful review. We agree that the paper will greatly benefit from editing to remove text redundancies and "wordiness, to better highlight the value of the datasets and the additional understanding we gained from it.
With respect to "reworking" the data: we are not entirely sure what the reviewer means. Obviously, we cannot "reanalyze" the data BUT we will revise the entire text to remove redundancies and partial "wordiness" in the manuscript. We are convinced this will result in a more "to the point" presentation of the results. We also would argue that the chosen analyses make the best use of the unique long-term dataset of stable water isotopes. Other studies of catchment inter-comparisons use isotopic datasets like ours and similar analyses (i.e. storm events, young water fractions, transit times) to study catchment behavior and assess the differential impacts of urbanization and/or climate change on discharge and catchment dynamics (i.e. Bonneau et al., 2018, von Freyberg et al., 2018). However, the wealth of our isotope data set is quite unique (in terms of length and resolution). Therefore, we would refrain from reworking any data, but rather improve the existing figures and text in such a way that they better present the insights gained from this study and highlights the uniqueness and usefulness of such long-term datasets.

1.  Focus
It is not clear from the introduction on and in the whole paper what the focus and objectives of the study really are. The Objectives section of the Introduction (l82-104) is very long and wordy.
** We will revise the introduction, shorten it and remove any redundant information.
Do the authors deal with seasonal patterns of flow response?
** yes.
Response to rainfall events?
** yes.
Response to climate change?

** no.

Recovery from drought events?

** No

What is the temporal scale of interest?

** we don't

Similarly, the title of the paper indicates that the main focus of the paper lies in isotope tracer results, but there is also a very long «classical» hydrologic analysis that is not very well articulated with the isotope sections. This lack of clear focus is really a problem when we come to results interpretation and conclusions.

** Analysing stable isotopes in hydrology only fully makes sense when conducted within a "general" hydrological analysis. The main novelty of this manuscript stems from the use of isotopes, as it gives context to understanding the different catchment responses.

However, we acknowledge the potential confusion and will condense the mentioned paragraph (L 82-104) to give a clearer outline of this study and its objectives.

That said, the focus of this paper was to firstly make use of the extensive datasets of daily stable water isotopes over multiple years and to study seasonal streamflow patterns of two (admittedly) contrasting catchments in the Berlin/Brandenburg region. This is addressing the scope of this SI, so the focus on drought was chosen to fit with the focus of the special issue, with the aim to understand the impact of hydroclimate forcing and anthropogenic water management. So, in a sense we are contrasting the extremes of heavily managed urban and agricultural extremes. Regarding temporal scale, our analysis is based on daily isotope and high-resolution discharge data over 5 years, focusing on seasonal dynamics. We will articulate this more clearly in the objectives section (temporal scale, key focus area) and also form a clearer hypothesis that guides the reader.

2.   Selection of the catchments

As far as I can see, the catchments are very different in all aspects: climate (although this part is not very clear), sizes (the urban catchment is much larger), land use of course, but also geologies. The urban catchment is also heavily managed, with water inflow from a WTTP and flood regulation (+ other minor unclear details, see detail remarks below). Are these catchments really comparable? What is the point of comparing them since they are so different? In the paper they are not really compared, the results are shown and discussed sequentially each time. It makes it really hard to draw general conclusions from this juxtaposed study and limits the impact of the paper.

** We agree that the sequential presentation of results may not be the most effective, and will edit the text in a way that uses more comparative language and also avoids repetition.

The catchments are both within 100km of each other and importantly, both are tributaries of the river Spree (with a catchment size of >10000 km2), which is a major water provider to the City of Berlin. Again, we would like to repeat that the focus of the Special issue where we submitted this paper to was on climate effects on water resources in the Berlin/Brandenburg region. Therefore, this study fits perfectly into the scope of this SI. We will make this clearer in the revision.

The catchments' regional climate / climate zone is therefore similar although the experiences differences in their local climate. Otherwise, in terms of their size, land use, geologies and management they are very different. But we chose this specific comparison as the urban catchment – while larger, did resemble the rural catchment in land use prior to the advanced urbanization. Our goal was to use these two contrasting catchments to understand baseflow responses following anthropogenic impact and extensive management, which is still somewhat underappreciated in hydrological studies.

We acknowledge that traditionally hydrological catchment comparisons tend to focus on catchments of similar size and characteristics, there has been plenty of previous international site comparison, sometimes spanning large environmental or climatic gradients, (i.e. Tetzlaff et al., 2009 a, b; von Freyberg, 2018) to assess differing catchment responses to climate forcing. Therefore, we believe that there is major value in the comparison of these two catchments, as it is precisely the juxtaposition of heavily managed urbanized and rural near natural streams environments, that are of interest in times of declining streamflow permanence and extreme events (droughts and extreme rainfall).

3. Methods

For the hydrological analysis, many indicators are mentioned, again with a confusion between seasonal patterns and response to storm events. We do not know which indicators were actually calculated, and we lack a few basic informations about typical orders of magnitude on the catchment to appreciate theses choices (eg how many events were selected, average characteristics, typical discharge values and so on). A table summarizing all the indicators that were actually calculated and for which objective would be very welcome.

** We acknowledge that there are quite a few indicators and parameters presented. Table 1 was meant to provide the necessary hydrological context to the annual differences in streamflow behavior (Q5, Q95, minQ, maxQ, baseflow index and runoff) between the two catchments.

Most of these indicators (i.e baseflow index, runoff coefficient, Q5 etc) are standard hydrological parameters that are calculated from the available data. We explain how they were calculated in the relevant method section (Section 3.1 Climate and hydrological data, L167 – 175).

We will provide an additional table summarizing the indicators and data its based and propose to include it in the supplementary material to avoid cluttering the main manuscript. This way, an interested reader can get the necessary information and background.

Regarding reference values and orders of magnitude, we will provide additional information in Section 2 of the study catchment description.

For the selection of rain events, this was also described in the section (L176 – 186). We will however, make sure the text is clearer on which parameters are calculated and which are measured to avoid confusion.

Is flow intermittence a topic of interest in the study? If yes, specific indicators could be looked at, plenty can be found in the literature. Same for «elasticity (l442). If the «recovery» from droughts is the main topic of interest (as stated in the paper's title), specific indicators can be also calculated (definition of drought events etc). I am not a specialist at all of isotope data, so I was not able to review specifically this section, but I would have appreciated a little more pedagogic explanations (perhaps with a schema explaining the various indicators calculated).

** The Berlin/Brandenburg region experiences increasingly stream intermittence. Our research group has published on this before, and we will briefly add this to the study site section citing the following papers - Luo et al., 2024; Ying et al., 2024; Kleine et al., 2021. Although we acknowledge the issues of intermittency in this paper, the regulation of the urban stream by waste water means that it is not directly an issue there, which is why we have not provided the metrics mentioned.

Also, recovery of drought is not the main focus. As stated above, the analysis revolves more around a general understanding of streamflow generation and response under temporally variable hydroclimate forcing, which included a drought period. Focusing only on drought responses would require a different kind of analysis and indeed different indicators and definition of drought periods/events etc. which is beyond the scope of this study (but was addressed in other studies by the group).

We realize the complexity of isotope data for the less experienced reader and appreciate the suggestion of additional explanations. However, we would like to point out that in the relevant method sections (3.2 and 3.3) we did provide extensive information regarding data collection and calculation of the different parameters (i.e. Local Meteoric Water line, lc-excess, water ages, transit times). We believe this information should give enough context and information for reproducibility and understanding. We respectfully disagree to provide an additional "schema" as this would not add any value to the interpretation or presentation of results. However, we will make sure to also edit the text in this section to be more succinct and focused for easy understanding.

**4. Results**

The Results section is very descriptive. The hydrology sections are lengthy stories of what happened in each catchment year after year, where a more synthetic analysis would have been expected. The Figures don't help. Figs 2-6 are extremely complex and contain way too much superimposed information, which is not necessary. For example in Fig 2, instead of presenting a full 5 year long hydrograph at 15 min time step that is completely illegible, it would have been much more interesting to present interannual flow regimes to study the seasonal patterns, and more focussed events for specific analyses. The authors also don't choose between comparing the different years and comparing the catchments. As a result, it is impossible to obtain a clear picture of what is going on.

**We realize there is a lot of information presented in the results section and appreciate the opportunity to revise it.

We will revise and shorten the entire result section to be more precise. Regarding discharge measurements presented in Fig. 1 - we will switch the presentation of 15min data to hourly data to make it more legible. We also realize that colors are not ideal and will change this to a more legible color scheme. We believe that showing the full 5 year hydrograph in relation to precipitation is important to provide a visual context of the different streamflow regimes and responses to climate forcing.

We present interannual perspectives on flow regimes in Figure 4 through the flow duration curves and discuss in Section 4.3 the seasonal patterns. Referee #2 suggested to add double mass curves to show cumulative precipitation and discharge, which we will do.

In response to similar comments by Referee 2, we will restructure the results section in such a way that we start with a general description of precipitation patterns and rain events. For this we will merge section 4.1 and 4.4 and reduce the text to present the most relevant results regarding the differences in the seasonal distribution of rainfall and dry periods, and the different storm events.

This would be followed by Section 4.2 – a description of the seasonal streamflow patterns.

Then Section 4.3 will deal with streamflow isotope patterns. Finally, Section 4.4. will present results regarding young water fractions and mean transit times.

We would argue that the isotope figures (Fig. 3, 5, 6) show an acceptable level of complexity similar to figures in other studies doing the same kind of analyses (i.e. papers cited in introduction, method and discussion sections).

We believe this way the results section will be more comprehensive and provide a better overview and understanding.

Some of the results in the text are also not supported by Figures, eg the section on storm events refers to the general hydrograph on Fig2 where nothing can be seen, and numeros correlations are mentioned in the text without supporting Figs or Tables.

**As we propose to merge the section on storm events (currently Section 4.4), this should clear up the concerns by the Referee. We will also make sure that any correlations will be referenced to the relevant Figures/Tables.

I was not able to review the Isotope sections but the corresponding Figures seem also very complicated and unclear to me (eg in Fig 3: I really don't see the differences between the catchments. For both there are points all over the place. More explanations are needed).

**We acknowledge that the symbols may be hard to distinguish in their current form. We will increase the size of the points to make them more visible. However, the representation of isotope results in dual isotope space (Fig.3) is a standard practice and meant to illustrate the variability and range of values found in each catchment. We will make sure that the text is clearer and more concise to avoid confusion and guide the reader through the figure and results. As a general interpretation: the closer the values are together, the less variable they are -meaning a more constant and similar water source is present in a stream, while points spread larger apart indicate greater variability in the source water contributions and seasonal variability.

**5. Discussion**

The discussion does not bring much in terms of interpretation of results, maybe because the results are so scattered. It is therefore a mix of descriptive talk and more general considerations that are not directly linked to the paper's subject (example blue / green water concepts) or partially repeat what was already said in the Introduction.

**We acknowledge the wordiness and "descriptive talk" and suggest to refocus the discussion to fit better with a redirected focus from the introduction.

We believe that our analyses do allow us to make a general link from the importance of understanding streamflow generation to blue/green infrastructure, especially in the urban environment, and we will write this clearer in the revised manuscript. We argue that first understanding streamflow dynamics in a catchment and understanding the ability of a catchment to store/release water is important to evaluate the effectiveness of such measures, especially in highly urbanized systems. We will clarify the novelty of such analyses – in particular for urban catchments. At the same time, this is also relevant in rural agricultural catchments where water bodies are increasingly important for maintaining blue-green fluxes and biodiversity.

Especially since streamflow generation and intermittency are becoming an increasingly important issue under advancing climate change (not just in the Berlin/Brandenburg region), we also believe it is relevant to highlight the use of stable water isotopes as a valuable tool to develop a more integrated understanding of hydrological dynamics, especially in ungauged basins where hydrometric data is less readily available.

Nevertheless, we will avoid repetition of arguments made in the introduction and also reframe the discussion sections to be more precise and better highlight the results and their implications.

The conceptual model in Fig 7 is a very good idea to sum up and present the conclusions of the study, but it lacks precision. Being too general, it fails to bring forward the results and show the knowledge added by the study. In its present state, it presents traditional hydrological processes, as can be found in any hydrology course and could have been guessed from the start.

** we will revise the figure as follows: we will add flux amounts in mm and water ages in months to link the figure more explicitly to the results. We will also aim to make land use a less prominent feature and focus more on the link to hydroclimate and highlight the aspect of urban water management.

**6. Detail remarks**

l260: is the drinking water for Berlin city withdrawn from the catchment? This part is not clear.

** Yes, water abstractions occur in the catchment. However, more water is imported into the catchment from the Spree and Havel, as Berlin depends on bank filtration to supply water to the city. We will reword the section to make this clearer.

l119: «flat lowland landscape»: is the only indication that we get about the topography. Is it possible to have a little more information, especially for the readers who are not familiar with the area?

**We will add more information about topography and elevation gradients.

Fig 1: the rural catchment is 60 km² but on the map the gauging station + sampling point is not located at the outlet, the catchment that was actually studied is much smaller then?

**Yes, the entire catchment is 60km$^2$ but since we are using the gauging station further up in the catchment – indeed the studied catchment area is slightly reduced. We will amend this in the description of the study site and insure this information is conveyed correctly to avoid misinterpretation and confusion.

p13: in the paragraph on seasonal flow regimes, there is a mention of response to precipitation events which is off topic + «evidenced by runoff coefficients»: where are these runoff coefficients? There is no ref to Fig or Table.

**The runoff coefficients were presented in Table 1 (Q/P). We will add reference to the correct Table to avoid confusion.

Fig 5: what are the grey lines?

**The grey lines in the plot have no specific meaning but were a graphical choice. They will be removed to simplify viewing of the plot.

**Additional References:**

Bonneau, Jeremie, et al. "The impact of urbanization on subsurface flow paths–A paired-catchment isotopic study." *Journal of Hydrology* 561 (2018): 413-426.

von Freyberg, J., Allen, S. T., Seeger, S., Weiler, M., & Kirchner, J. W. (2018). Sensitivity of young water fractions to hydro-climatic forcing and landscape properties across 22 Swiss catchments. *Hydrology and Earth System Sciences*, *22*(7), 3841-3861.

Kleine L, Tetzlaff D, Smith A, Goldhammer T, Soulsby C. (2021) Using isotopes to understand landscape-scale connectivity in a groundwater-dominated, lowland catchment under drought conditions. *Hydrological Processes*. http://dx.doi.org/10.1002/hyp.14197

Luo S, Tetzlaff D, Smith A, Soulsby C. (2024) Long-term drought effects on landscape water storage and resilience under contrasting landuses. *Journal of Hydrology*, https://doi.org/10.1016/j.jhydrol.2024.131339

Marx, C., Tetzlaff, D., Hinkelmann, R., & Soulsby, C. (2021). Isotope hydrology and water sources in a heavily urbanized stream.*Hydrological Processes*,*35*(10), e14377.

Tetzlaff D, Seibert J, Soulsby C. (2009) Inter-catchment comparison to assess the influence of topography and soils on catchment transit times in a geomorphic province; the Cairngorm Mountains, Scotland. *Hydrological Processes*, 23, 1874–1886.

Tetzlaff D, Seibert J, McGuire KJ, Laudon H, Burns DA, Dunn SM, Soulsby C. (2009) How does landscape structure influence catchment transit times across different geomorphic provinces? *Hydrological Processes* 23, 945–953

Ying Z, Tetzlaff D, Freymueller J, Comte JC, Goldhammer T, Schmidt A, Soulsby C (2024) Developing a conceptual model of groundwater – surface water interactions in a drought sensitive lowland catchment using multi-proxy data. *Journal of Hydrology*, https://doi.org/10.1016/j.jhydrol.2023.130550

---

## Author Comment (AC2)

**Reply to Comments by Referee #2:**
Dear Referee,
Thank you for giving us the opportunity to revise our manuscript. We appreciate the careful review and the comments and suggestions provided. We believe these comments will help to strengthen the focus of this paper and improve the message and key points we are trying to convey. Below, we address the specific comments as they were made, point by point and provide clarifications where necessary. We are confident that through this process we can improve the structure and effectiveness of the paper and communicate the results more clearly.
Sincerely,
Dr. Maria Magdalena Warter (on behalf of all co-authors)

Reply to General Comment:
The authors of this manuscript carried out an inter-comparison study of two anthropogenically impacted catchments (rural vs. urban land use), by integrating a hydro-meteorological and an isotopic-based monitoring. Data used for the analysis cover about five hydrological years, and such high-resolution isotopic datasets are particularly rare, especially for urban catchments. These datasets were used to investigate how drought periods affect the hydrological functioning of the two catchments and to characterize runoff persistence and resilience during droughts and in response to storm events.
The topic of the manuscript falls within the scope of the journal, and this study could represent a valuable contribution. Overall, the paper is well structured and written, but I have some major concerns that should be addressed in the revision. First of all, based on the discussion, it seems that most of the differences in the hydrological functioning of the two catchments is related to the very different land use; however, the inter-comparison was not conducted on two catchments with just a different land use, because they also differ in area, geology and annual rainfall. Secondly, based on Figure 1, it looks like that the density of weather stations is very low considering the size of the two catchments, and therefore, I am wondering whether rainfall measurements (especially during storm events) are representative of the entire catchments. Finally, I think that at the beginning of the results there should be a section focusing only on the seasonal distribution of the rainfall, the characterization of the drought periods as well as on the storm events (something described later in Section 4.4).

** First, we would like to thank Referee 2 for their overall positive evaluation and also their critical feedback.
We noticed that we weren't clear in our original manuscript re that both catchments are tributaries of the river Spree, a major river for the water supply of the City of Berlin. Thus, they are located in the same regional climate zone though do show different local climates.
The two presented catchments are quite different in their land use, size, and geology. However, as also mentioned in reply to Referee 1, there are similar studies that conducted such inter-comparisons on hydrological responses of catchments that differ in size, underlying geology and hydroclimate properties (i.e. Tetzlaff et al., 2009a, b; von Freyberg et al. 2018). Our goal was to do something similar by using these admittedly contrasting catchments to understand how two key endmembers (urban vs agricultural) of anthropogenically impacted catchments, which are climatically impacted in Berlin/Brandenburg region, which was the focus of the special issue that this manuscript was submitted to.
However, we would also like to note that while current land use in both catchments may be different now, the urban catchment had a similarly agriculturally dominated land use prior to the rapid expansion of urban areas. Therefore, we believe that comparing these two specific catchments allows us to also evaluate in a way the effects of urbanization and streamflow management on streamflow generation in times of drought and extreme events, compared to rural less managed streams.

Secondly, regarding weather stations, we primarily used open source long-term data, and their number is limited. The station in Berlin Buch (open data by the German Weather Service) has been used in previous studies by the group of the Panke catchment (see Marx et al., 2021, 2023) and is representative for the catchment. The distance between weather station and catchment outlet is <15km. Similarly, the weather station in Hasenfelde (Brandenburg) has been used in previous studies of the Demnitzer Millcreek catchment (see e.g. Kleine et al., 2020, 2021) and is considered to be representative of rainfall dynamics (distance < 10km) in the area. As the focus of the study is not detailed storm event analysis we would argue that the use of these stations for the scope of our study is acceptable.

Finally, we agree that the results section will benefit from editing and appreciate the suggestion to restructure it. In line with similar suggestions from Referee 1, we will start with a presentation of the seasonal distribution of rainfall and rainfall events and also highlight more the dry periods in between. We will do this by merging text from sections 4.1 and 4.3 into a more condensed form to reduce wordiness and repetition.

This will be followed by a description of the streamflow patterns (Section 4.2) and isotope dynamics (Section 4.3) and finally the description of young water fractions and transit times (Section 4.4).

**Specific comments:**

Section 2: These two catchments have more differences than similarities, so I am not sure that many findings can be related mostly to the land use. Maybe the focus of the manuscript should be more on the analysis of inter-annual variability (and on droughts) than on the catchment inter-comparison.

** We were not clear enough in our original submission that both catchments are actually located in close proximity (ca. 100 km) and both tributaries of one major river system (the Spree). We appreciate the suggestion to focus more on a comparative analysis of inter-annual variability of streamflow generation and the expression of drought. We will give the drought more emphasis in the revision to fit also with the topic of the SI (drought risks in Berlin/Brandenburg region).

Figure 1: There are very few weather stations in the two study areas; are the rainfall measurements representative of the real spatio-temporal variability of rainfall over the entire catchments? Did the authors check the measurements during storm events and compare them to weather radar data?

**Yes, as mentioned above the two weather stations can be considered representative of the two catchments and have been regularly used in previous studies in the same catchments (see Marx et al., 2021, 2022, Kleine et al., 2021, 2020). We are therefore confident that using the two weather stations sufficiently captures the spatio-temporal variability of rainfall over each respective catchment. We point out that we are not modeling at sub-daily time steps, where convectional differences would be more important and require a higher resolution of weather data.

Figure 4a: Despite the different land use, area and geologies, for WY2019 I was expecting to see the lowest discharges in both catchments (compared to the following years). Based on the flow duration curves, it is clear that the different climatic conditions in the two catchments may have led to a different runoff response.

**Rather than only different climatic conditions, this is also a result of increased contributions of effluent into the urban catchment during the drought, that causes the increased discharge in the urban stream. Furthermore, in the urban area of Berlin during WY 2018/19 there were still several large summer convective events (up to ~50mm) while in the rural area, no rainfall was

recorded for several weeks between March – May and only limited rainfall in summer, resulting in a much more severe decrease in streamflow.

The effects of the drought only became fully visible in WY 2019/20 in the urban area – as seen by the lowest discharges in that year (compared to following years).

When plotting the double mass curves (see below) per WY, the differences in cumulative amounts become even clearer between WY2019 and the following WY2020 (see below), with the effects of drought being visible in WY 2020.

Section 4.3: Besides flow duration curves, I recommend adding double-mass curves (cumulative precipitation vs. cumulative specific discharge) for comparing the hydrological response of the two catchments during different years and at the seasonal scale (the focus could be on drought periods as well as on very wet months).

**We appreciate the suggestion to add double-mass curves for comparing the hydrological responses and will do so during the review. We will add them to Figure 4 to complement the flow duration curves.

Section 4.4: The characterization of the rainfall events should be anticipated in the results and merged to the first section of the results, in order to help understand the hydrographs and the flow duration curves.

**We agree with this suggestion and will restructure the section to present the seasonal distribution as well as characterization of rainfall events at the beginning of the result section, followed then by the results of streamflow patterns and flow duration.

Section 4.5: What is the sensitivity of young water fraction estimates on the sampling design for both rainfall (how many collectors were used?) and stream water? I wonder whether capturing the isotopic variability during flashy events in the urban catchment would have determined a different estimation. Furthermore, in this case, results on young water fractions and MTT may be due to a combination of factors, such as catchment area, geology and land use.

**For the collection of rainfall isotopes, generally only 1 collector is used. We did use two separate datasets of precipitation isotopes from Berlin Steglitz (for the urban catchment) and from the AWS in Hasenfelde (for the rural catchment).

Regarding sampling of stream water isotopes, we collected daily samples to insure continuity. This is a high resolution for stable isotopes in particular as sampled of longer periods (inter-annually). However, it is likely that sampling over the course of a rain event could give different estimates of young water contributions, which may be more damped when sampling just on daily basis, especially in the urban catchment where runoff responses can be relatively quick. From previous sampling, we know that there is minimal effect on young water estimates during flow peaks.

We agree, that estimates of young water fractions and MTT are likely due to a combination of factors related to land use and catchment area, and in the case of the urban catchment – the overwhelming influence of wastewater, which complicates the estimation of MTTs.

Table 2: Adjusted R2 are very low and RSE are very high for the urban catchment. Perhaps, these values should be considered carefully during the interpretation of the results.

**Yes, we agree these values need to be considered with care. They highlight the difficulties of these methods to estimate young water fractions in urban catchments with a strong influence of wastewater, as the isotopic variability is much more damped than in the rural stream, resulting in higher RSE and lower R2. We will make sure to critically discuss this in the discussion section.

Section 5.3 should be revised (please see the comment about the inter-comparison); based on this text and Figure 7, it seems that land use represents the main factor determining the different hydrological functioning of the two catchments during drought and wet periods.
**We agree that the discussion about the wider implications will benefit from revision and more specific arguments. As mentioned also to Referee 1, we will amend the schematic graphic in such a way that there is less focus on land use effects but on the aspects of different streamflow patterns under drought/wet periods in different anthropogenic environments. We will add amounts of young water estimates (in months) and flux amounts (in mm) to give a specific link to the results.

Figure S2: In late April of WY2020, there should be a rainfall event triggering the flash and marked discharge increase; however, there is only a rainfall pulse with a very low magnitude. Is this correct? Discharge in WY2020 seems to have a very different behaviour compared to the following years; is there a specific explanation?
**These distinct flashes in the urban streamflow regime are not necessarily associated with rainfall pulses but rather with streamflow management and opening of a weir upstream. This generally triggers such a marked streamflow response downstream, as is visible between late April and May 2020. This is usually done during drier periods – as was the case during the spring of 2020, to increase the flowrate and avoid the stream drying out. In this particular case the increased flow lasted for about 2,5 weeks before being reduced again (weir closed). We will make clear in the revised manuscript that this context is understood.

**Technical corrections**
Line 84: 'as part of' can be deleted.
**Agreed. Will be corrected.

Line 96: 'selected' instead of 'select'.
**Agreed. Will be corrected.

Line 100: 'in an integrated way' can be deleted.
**Agreed. Will be corrected.

Line 216: 'MTT' instead of 'MMT'.
**Will be changed.

Figure 2: For baseflow I see pink or purple lines, not red lines. Furthermore, I do not see the winter and summer events highlighted in red and green, respectively.
**We will change the color scheme to be more visible and easier to distinguish.

Line 398: If the correlation is negative, r should be -0.49.
**Yes, will be changed.

Line 434: '2018-19' instead of '208-19'.
**Will be changed.

Line 499: 'due to' repeated twice.
**Will be deleted.

Line 500: Unclear 'lack of recharge. Hydromorphic conditions…'.
**The sentence will be clarified – a connecting phrase was missing.

**Additional References:**

Bonneau, Jeremie, et al. "The impact of urbanization on subsurface flow paths–A paired-catchment isotopic study." *Journal of Hydrology* 561 (2018): 413-426.

von Freyberg, J., Allen, S. T., Seeger, S., Weiler, M., & Kirchner, J. W. (2018). Sensitivity of young water fractions to hydro-climatic forcing and landscape properties across 22 Swiss catchments.*Hydrology and Earth System Sciences*, *22*(7), 3841-3861.

Kleine L, Tetzlaff D, Smith A, Goldhammer T, Soulsby C. (2021) Using isotopes to understand landscape-scale connectivity in a groundwater-dominated, lowland catchment under drought conditions. *Hydrological Processes.* http://dx.doi.org/10.1002/hyp.14197

Kleine, L., Tetzlaff, D., Smith, A., Wang, H., & Soulsby, C. (2020). Using water stable isotopes to understand evaporation, moisture stress, and re-wetting in catchment forest and grassland soils of the summer drought of 2018. *Hydrology and Earth System Sciences*, *24*(7), 3737-3752.

Marx, C., Tetzlaff, D., Hinkelmann, R., & Soulsby, C. (2021). Isotope hydrology and water sources in a heavily urbanized stream.*Hydrological Processes*,*35*(10), e14377.

Marx, C., Tetzlaff, D., Hinkelmann, R., & Soulsby, C. (2023). Effects of 66 years of water management and hydroclimatic change on the urban hydrology and water quality of the Panke catchment, Berlin, Germany. *Science of the Total Environment*, *900*, 165764.

Tetzlaff D, Seibert J, Soulsby C. (2009) Inter-catchment comparison to assess the influence of topography and soils on catchment transit times in a geomorphic province; the Cairngorm Mountains, Scotland. *Hydrological Processes*, 23, 1874–1886.

Tetzlaff D, Seibert J, McGuire KJ, Laudon H, Burns DA, Dunn SM, Soulsby C. (2009) How does landscape structure influence catchment transit times across different geomorphic provinces? *Hydrological Processes* 23, 945–953

---

## Author Response (AR1)

**Response to Referee Comment 1:**

Thank you for your constructive comments and suggestions. We believe addressing these comments will strengthen the paper and improve the message and key points we are trying to convey. Below, we respond to the specific comments, point by point and provide clarifications where necessary. We are confident that through this process we can improve the structure and effectiveness of the paper and communicate the results more clearly. Importantly, we are not sure if the reviewer was made aware that this paper was submitted to a Special Issue on the region "Berlin/Brandenburg" with a very specific local focus. Considering the scope of the SI might help to clarify some of the comments made.

Sincerely,

Dr. Maria Magdalena Warter (on behalf of all co-authors)
* * *
This paper by Warter et al. deals with the resilience of streams facing droughts. This is a very interesting topic, as this resilience is due to complex processes that are dependent on interacting catchment characteristics (climate, geology, pedology, land use, water management practices,…). The study analyses hydrological and stable water isotopes data from a 5-year data set on 2 contrasted catchments in Germany (sizes, geologies, land uses,…).

The paper clearly has a lot of potential and deals with a large amount of data.

** Thank you for this positive evaluation of our manuscript.

However, I found it very long, wordy, and difficult to read, mainly because it lacks focus and precision in the analysis. Therefore, it is complicated for the reader to appreciate the results and the impact of the paper. My suggestion would be to rework the data to be able to present less «raw» and more to the point results. My main remarks and recommendations are listed below:

**We thank the reviewer for the careful review. We agree that the paper will greatly benefit from editing to remove text redundancies and "wordiness, to better highlight the value of the datasets and the additional understanding we gained from it.

With respect to "reworking" the data: we are not entirely sure what the reviewer means. Obviously, we cannot "reanalyze" the data BUT we revised the entire text to remove redundancies and partial "wordiness" in the manuscript. We are convinced this resulted in a more "to the point" presentation of the results. We also would argue that the chosen analyses make the best use of the unique long-term dataset of stable water isotopes. Other studies of catchment inter-comparisons use isotopic datasets like ours and similar analyses (i.e. storm events, young water fractions, transit times), to study catchment behavior and assess the differential impacts of urbanization and/or climate change on discharge and catchment dynamics (i.e. Bonneau et al., 2018, von Freyberg et al., 2018). However, the wealth of our isotope data set is quite unique (in terms of length and resolution). Therefore, we refrained from reworking any data, but rather improved the existing figures and text in such a way that they better present the insights gained from this study and highlight the uniqueness and usefulness of such long-term datasets.

1. Focus

It is not clear from the introduction on and in the whole paper what the focus and objectives of the study really are. The Objectives section of the Introduction (l82-104) is very long and wordy.

** We revised the introduction and removed any redundant information.

Do the authors deal with seasonal patterns of flow response?

** yes.

Response to rainfall events?

** yes.

Response to climate change?

** no.

Recovery from drought events?
** No
What is the temporal scale of interest?
** We worked with daily stable isotope data and hourly discharge/precipitation data to analyze streamflow and storm responses.

Similarly, the title of the paper indicates that the main focus of the paper lies in isotope tracer results, but there is also a very long «classical» hydrologic analysis that is not very well articulated with the isotope sections. This lack of clear focus is really a problem when we come to results interpretation and conclusions.
** Analysing stable isotopes in hydrology only fully makes sense when conducted within a "general" hydrological analysis. The main novelty of this manuscript stems from the use of isotopes, as it gives context to understanding the different catchment responses.
We condensed the mentioned paragraph (L 82-104) to give a clearer outline of this study and its objectives.

We articulated the focus of the paper more clearly in the objectives section (temporal scale, key focus area) and also presented clearer objectives.

That said, the focus of this paper was to firstly make use of the extensive datasets of daily stable water isotopes over multiple years and to study seasonal streamflow patterns of two (admittedly) contrasting catchments in the Berlin/Brandenburg region. This is addressing the scope of this SI, so the focus on drought was chosen to fit with the focus of the special issue, with the aim to understand the impact of hydroclimate forcing and anthropogenic water management on streamflow generation. So, in a sense we are contrasting the extremes of heavily managed urban and agricultural extremes to understand streamflow generation. Regarding temporal scale, our analysis was based on daily isotope and high-resolution discharge data over 5 years, but focusing on seasonal dynamics.

2. Selection of the catchments
As far as I can see, the catchments are very different in all aspects: climate (although this part is not very clear), sizes (the urban catchment is much larger), land use of course, but also geologies. The urban catchment is also heavily managed, with water inflow from a WTTP and flood regulation (+ other minor unclear details, see detail remarks below). Are these catchments really comparable? What is the point of comparing them since they are so different? In the paper they are not really compared, the results are shown and discussed sequentially each time. It makes it really hard to draw general conclusions from this juxtaposed study and limits the impact of the paper.
** We edited the text in a way that uses more comparative language and also avoids repetition. The catchments are both within 100km of each other and importantly, both are tributaries of the river Spree (with a catchment size of >10000 km2), which is a major water provider to the City of Berlin. Again, we would like to repeat that the focus of the Special issue where we submitted this paper to was on climate effects on water resources in the Berlin/Brandenburg region. Therefore, this study fits perfectly into the scope of this SI. We made this clearer in the introduction.

The catchments' regional climate / climate zone is therefore similar although they experience differences in their local climate. Otherwise, in terms of their size, land use, geologies and management they are very different. But we chose this specific comparison as the urban catchment – while larger, did resemble the rural catchment in land use prior to the advanced

urbanization. Our goal was to use these two contrasting catchments to understand baseflow responses following anthropogenic impact and extensive management, which is still somewhat underappreciated in hydrological studies.

We acknowledge that traditionally hydrological catchment comparisons tend to focus on catchments of similar size and characteristics, there has been plenty of previous international site comparison, sometimes spanning large environmental or climatic gradients, (i.e. Tetzlaff et al., 2009 a, b; von Freyberg, 2018) to assess differing catchment responses to climate forcing. Therefore, we believe that there is major value in the comparison of these two catchments, as it is precisely the juxtaposition of heavily managed urbanized and rural near natural streams environments, that are of interest in times of declining streamflow permanence and extreme events (droughts and extreme rainfall).

**3. Methods**

For the hydrological analysis, many indicators are mentioned, again with a confusion between seasonal patterns and response to storm events. We do not know which indicators were actually calculated, and we lack a few basic informations about typical orders of magnitude on the catchment to appreciate theses choices (eg how many events were selected, average characteristics, typical discharge values and so on). A table summarizing all the indicators that were actually calculated and for which objective would be very welcome.

** We acknowledge that there are quite a few indicators and parameters presented. Table 1 was meant to provide the necessary hydrological context to the annual differences in streamflow behavior (Q5, Q95, minQ, maxQ, baseflow index and runoff) between the two catchments.

Most of these indicators (i.e baseflow index, runoff coefficient, Q5 etc) are standard hydrological parameters that are calculated from the available data. We explain how they were calculated in the relevant method section (Section 3.1 Climate and hydrological data, L167 – 175).

We provided an additional table summarizing the indicators and data its based on and included it in the supplementary material (Table S2) to avoid cluttering the main manuscript. This way, an interested reader can get the necessary information and background.

Regarding reference values and orders of magnitude, we provided additional values in Section 2 of the study catchment description.

For the selection of rain events, this is described in the method section (L176 – 186). We made the text clearer on which parameters are calculated and which are measured.

Is flow intermittence a topic of interest in the study? If yes, specific indicators could be looked at, plenty can be found in the literature. Same for «elasticity (l442). If the «recovery» from droughts is the main topic of interest (as stated in the paper's title), specific indicators can be also calculated (definition of drought events etc). I am not a specialist at all of isotope data, so I was not able to review specifically this section, but I would have appreciated a little more pedagogic explanations (perhaps with a schema explaining the various indicators calculated).

** The Berlin/Brandenburg region increasingly experiences stream intermittence. Our research group has published on this before, and we added relevant reference in the introduction (L86) citing the following papers - Luo et al., 2024; Ying et al., 2024; Kleine et al., 2021.

Although we acknowledge the issues of intermittency in this paper, the regulation of the urban stream by waste water means that it is not directly an issue there, which is why we have not provided the metrics mentioned.

Also, recovery of drought is not the main focus. As stated above, the analysis revolves more around a general understanding of streamflow generation and response under temporally variable hydroclimate forcing, which included a drought period. Focusing only on drought responses would require a different kind of analysis and indeed different indicators and definition of drought periods/events etc. which is beyond the scope of this study (but was addressed in other studies by the group).

We realize the complexity of isotope data for the less experienced reader and appreciate the suggestion of additional explanations. However, we would like to point out that in the relevant method sections (3.2 and 3.3) we already provided extensive information regarding data collection and calculation of the different parameters (i.e. Local Meteoric Water line, lc-excess, water ages, transit times). We believe this information gives enough context and information for reproducibility and understanding. We respectfully disagree to provide an additional "schema" as this would not add any value to the interpretation or presentation of results. However, we edited the text in this section to be more focused for easy understanding.

4. Results

The Results section is very descriptive. The hydrology sections are lengthy stories of what happened in each catchment year after year, where a more synthetic analysis would have been expected. The Figures don't help. Figs 2-6 are extremely complex and contain way too much superimposed information, which is not necessary. For example in Fig 2, instead of presenting a full 5 year long hydrograph at 15 min time step that is completely illegible, it would have been much more interesting to present interannual flow regimes to study the seasonal patterns, and more focussed events for specific analyses. The authors also don't choose between comparing the different years and comparing the catchments. As a result, it is impossible to obtain a clear picture of what is going on.

** We changed the colors in Fig. 1 for better readability and amended the Figure caption. We used hourly normalized specific discharge for better comparison instead of 15 min streamflow. We believe that showing the full 5 year hydrograph in relation to precipitation is important to provide a visual context of the different streamflow regimes and responses to climate forcing.

We added double mass curves to Figure 3 (former Figure 4) to show cumulative precipitation and discharge.

We restructured the results section in such a way that we start with a general description of precipitation patterns and rain events. We merged section 4.1 and 4.4 and reduced the text to present the most relevant results regarding the differences in the seasonal distribution of rainfall and dry periods, and the different storm events.

This is followed by Section 4.2 – a description of the seasonal streamflow patterns.

Then Section 4.3 presents streamflow isotope patterns. Finally, Section 4.4. presents results regarding young water fractions and mean transit times.

We would argue that the isotope figures (Fig. 4 (former Fig. 3), 5, 6) show an acceptable level of complexity similar to figures in other studies doing the same kind of analyses (i.e. papers cited in introduction, method and discussion sections) and have left the figures as is.

Some of the results in the text are also not supported by Figures, eg the section on storm events refers to the general hydrograph on Fig2 where nothing can be seen, and numerous correlations are mentioned in the text without supporting Figs or Tables.

\*\*We merged the section on storm events (was Section 4.4), and made sure that any correlations were referenced to the relevant Figures/Tables.

I was not able to review the Isotope sections but the corresponding Figures seem also very complicated and unclear to me (eg in Fig 3: I really don't see the differences between the catchments. For both there are points all over the place. More explanations are needed).
\*\*We acknowledge that the symbols may be hard to distinguish in their current form. We increased the size of the points to make them more visible. However, the representation of isotope results in dual isotope space as shown in Figure 4, (former Fig. 3) is a standard practice and meant to illustrate the variability and range of values found in each catchment.

We condensed the text to be clearer and more concise (L411-422)

As a general interpretation: the closer the values are together, the less variable they are -meaning a more constant and similar water source is present in a stream, while points spread larger apart indicate greater variability in the source water contributions and seasonal variability.

**5. Discussion**
The discussion does not bring much in terms of interpretation of results, maybe because the results are so scattered. It is therefore a mix of descriptive talk and more general considerations that are not directly linked to the paper's subject (example blue / green water concepts) or partially repeat what was already said in the Introduction.
\*\*We acknowledge the wordiness and "descriptive talk" and reduced some text in the discussion.

However, we believe that our analyses do allow us to make a general link from the importance of understanding streamflow generation to blue/green infrastructure, especially in the urban environment, and we made this clearer in the revised manuscript. We argue that first understanding streamflow dynamics in a catchment and understanding the ability of a catchment to store/release water is important to evaluate the effectiveness of such measures, especially in highly urbanized systems. We clarified the novelty of such analyses – in particular for urban catchments. At the same time, this is also relevant in rural agricultural catchments where water bodies are increasingly important for maintaining blue-green fluxes and biodiversity.
Especially since streamflow generation and intermittency are becoming an increasingly important issue under advancing climate change (not just in the Berlin/Brandenburg region), we also believe it is relevant to highlight the use of stable water isotopes as a valuable tool to develop a more integrated understanding of hydrological dynamics, especially in ungauged basins where hydrometric data is less readily available.

Nevertheless, we reworded some of the text in the discussion to be more precise and highlight the results and their implications, without repeating results already presented. .

The conceptual model in Fig 7 is a very good idea to sum up and present the conclusions of the study, but it lacks precision. Being too general, it fails to bring forward the results and show the knowledge added by the study. In its present state, it presents traditional hydrological processes, as can be found in any hydrology course and could have been guessed from the start.
\*\* We revised the figure as follows: we added flux amounts in mm and % and water ages in days as well as lc-excess to link the figure more explicitly to the results. We also made land use a less prominent feature and focused more on the link to hydroclimate and the impact of urban water management on streamflow generation.

**6. Detail remarks**

l260: is the drinking water for Berlin city withdrawn from the catchment? This part is not clear.

\*\* Yes, water abstractions occur in the catchment. However, more water is imported into the catchment from the Spree and Havel, as Berlin depends on bank filtration to supply water to the city. We added this in the study catchment description.

l119: «flat lowland landscape»: is the only indication that we get about the topography. Is it possible to have a little more information, especially for the readers who are not familiar with the area?

\*\*Additional information regarding topography was provided in the study catchment description. (L136-138)

Fig 1: the rural catchment is 60 km² but on the map the gauging station + sampling point is not located at the outlet, the catchment that was actually studied is much smaller then?

\*\*Yes, the entire catchment is 60km$^2$ but since we are using the gauging station further up in the catchment – indeed the studied catchment area is slightly reduced (42 km$^2$). We added this in the description of the study site to insure this information is conveyed correctly.

p13: in the paragraph on seasonal flow regimes, there is a mention of response to precipitation events which is off topic + «evidenced by runoff coefficients»: where are these runoff coefficients? There is no ref to Fig or Table.

\*\*The runoff coefficients were presented in Table 1 (Q/P). Added reference in the text.

Fig 5: what are the grey lines?

\*\*The grey lines in the plot have been removed.

**Additional References:**

Bonneau, Jeremie, et al. "The impact of urbanization on subsurface flow paths–A paired-catchment isotopic study." *Journal of Hydrology* 561 (2018): 413-426.

von Freyberg, J., Allen, S. T., Seeger, S., Weiler, M., & Kirchner, J. W. (2018). Sensitivity of young water fractions to hydro-climatic forcing and landscape properties across 22 Swiss catchments. *Hydrology and Earth System Sciences*, 22(7), 3841-3861.

Kleine L, Tetzlaff D, Smith A, Goldhammer T, Soulsby C. (2021) Using isotopes to understand landscape-scale connectivity in a groundwater-dominated, lowland catchment under drought conditions. *Hydrological Processes.* http://dx.doi.org/10.1002/hyp.14197

Luo S, Tetzlaff D, Smith A, Soulsby C. (2024) Long-term drought effects on landscape water storage and resilience under contrasting landuses. *Journal of Hydrology*, https://doi.org/10.1016/j.jhydrol.2024.131339

Marx, C., Tetzlaff, D., Hinkelmann, R., & Soulsby, C. (2021). Isotope hydrology and water sources in a heavily urbanized stream.*Hydrological Processes*,35(10), e14377.

Tetzlaff D, Seibert J, Soulsby C. (2009) Inter-catchment comparison to assess the influence of topography and soils on catchment transit times in a geomorphic province; the Cairngorm Mountains, Scotland. *Hydrological Processes*, 23, 1874–1886.

Tetzlaff D, Seibert J, McGuire KJ, Laudon H, Burns DA, Dunn SM, Soulsby C. (2009) How does landscape structure influence catchment transit times across different geomorphic provinces? *Hydrological Processes* 23, 945–953

Ying Z, Tetzlaff D, Freymueller J, Comte JC, Goldhammer T, Schmidt A, Soulsby C (2024) Developing a conceptual model of groundwater – surface water interactions in a drought sensitive lowland catchment using multi-proxy data. *Journal of Hydrology*, https://doi.org/10.1016/j.jhydrol.2023.130550

Dear Referee,

Thank you for giving us the opportunity to revise our manuscript. We appreciate the careful review and the comments and suggestions provided. We believe these comments have helped to strengthen the focus of this paper and improve the message and key points we are trying to convey. Below, we address the specific comments as they were made, point by point and provide clarifications where necessary. We are confident that through this process we can improve the structure and effectiveness of the paper and communicate the results more clearly.

Sincerely,

Dr. Maria Magdalena Warter (on behalf of all co-authors)

Reply to General Comment:

The authors of this manuscript carried out an inter-comparison study of two anthropogenically impacted catchments (rural vs. urban land use), by integrating a hydro-meteorological and an isotopic-based monitoring. Data used for the analysis cover about five hydrological years, and such high-resolution isotopic datasets are particularly rare, especially for urban catchments. These datasets were used to investigate how drought periods affect the hydrological functioning of the two catchments and to characterize runoff persistence and resilience during droughts and in response to storm events.

The topic of the manuscript falls within the scope of the journal, and this study could represent a valuable contribution. Overall, the paper is well structured and written, but I have some major concerns that should be addressed in the revision. First of all, based on the discussion, it seems that most of the differences in the hydrological functioning of the two catchments is related to the very different land use; however, the inter-comparison was not conducted on two catchments with just a different land use, because they also differ in area, geology and annual rainfall. Secondly, based on Figure 1, it looks like that the density of weather stations is very low considering the size of the two catchments, and therefore, I am wondering whether rainfall measurements (especially during storm events) are representative of the entire catchments. Finally, I think that at the beginning of the results there should be a section focusing only on the seasonal distribution of the rainfall, the characterization of the drought periods as well as on the storm events (something described later in Section 4.4).

** First, we would like to thank Referee 2 for their overall positive evaluation and also their critical feedback.

We noticed that we weren't clear in our original manuscript re that both catchments are tributaries of the river Spree, a major river for the water supply of the City of Berlin. Thus, they are located in the same regional climate zone though do show different local climates. We have now included this information in section 2.2.

The two presented catchments are quite different in their land use, size, and geology. However, as also mentioned in reply to Referee 1, there are similar studies that conducted such inter-comparisons on hydrological responses of catchments that differ in size, underlying geology and hydroclimate properties (i.e. Tetzlaff et al., 2009a, b; von Freyberg et al. 2018). Our goal was to do something similar by using these admittedly contrasting catchments to understand how two key endmembers (urban vs agricultural) of anthropogenically impacted catchments, which are climatically impacted in Berlin/Brandenburg region, which was the focus of the special issue that this manuscript was submitted to.

However, we would also like to note that while current land use in both catchments may be different now, the urban catchment had a similarly agriculturally dominated land use prior to the rapid expansion of urban areas. Therefore, we believe that comparing these two specific catchments allows us to also evaluate in a way the effects of urbanization and streamflow

management on streamflow generation in times of drought and extreme events, compared to rural less managed streams.

Secondly, regarding weather stations, we primarily used open source long-term data, and their number is limited. The station in Berlin Buch (open data by the German Weather Service) has been used in previous studies by the group of the Panke catchment (see Marx et al., 2021, 2023) and is representative for the catchment. The distance between weather station and catchment outlet is <15km. Similarly, the weather station in Hasenfelde (Brandenburg) has been used in previous studies of the Demnitzer Millcreek catchment (see e.g. Kleine et al., 2020, 2021) and is considered to be representative of rainfall dynamics (distance < 10km) in the area. As the focus of the study is not detailed storm event analysis we would argue that the use of these stations for the scope of our study is acceptable.

We edited the results section to make this clearer. In line with similar suggestions from Referee 1, we started with the presentation of the seasonal distribution of rainfall and responses to storm events, but also highlighting the dry periods in between. We merged text from sections 4.1 and 4.3 and shortened it. This is now followed by a description of the streamflow patterns (Section 4.2) and isotope dynamics (Section 4.3) and finally the description of young water fractions and transit times (Section 4.4).

**Specific comments:**
Section 2: These two catchments have more differences than similarities, so I am not sure that many findings can be related mostly to the land use. Maybe the focus of the manuscript should be more on the analysis of inter-annual variability (and on droughts) than on the catchment inter-comparison.
** We were not clear enough in our original submission that both catchments are actually located in close proximity (ca. 100 km) and both tributaries of one major river system (the Spree). We appreciate the suggestion to focus more on a comparative analysis of inter-annual variability of streamflow generation and the expression of drought. We gave the drought more emphasis during the revision, in line with the topic of the SI (drought risks in Berlin/Brandenburg region).

Figure 1: There are very few weather stations in the two study areas; are the rainfall measurements representative of the real spatio-temporal variability of rainfall over the entire catchments? Did the authors check the measurements during storm events and compare them to weather radar data?
**Yes, as mentioned above the two weather stations can be considered representative of the two catchments and have been regularly used in previous studies in the same catchments (see Marx et al., 2021, 2022, Kleine et al., 2021, 2020). We are therefore confident that using the two weather stations sufficiently captures the spatio-temporal variability of rainfall over each respective catchment. We point out that we are not modeling at sub-daily time steps, where convectional differences would be more important and require a higher resolution of weather data.

Figure 4a: Despite the different land use, area and geologies, for WY2019 I was expecting to see the lowest discharges in both catchments (compared to the following years). Based on the flow duration curves, it is clear that the different climatic conditions in the two catchments may have led to a different runoff response.
**Rather than only different climatic conditions, this is also a result of increased contributions of effluent into the urban catchment during the drought, that causes the increased discharge in the urban stream. Furthermore, in the urban area of Berlin during WY 2018/19 there were still

several large summer convective events (up to ~50mm) while in the rural area, no rainfall was recorded for several weeks between March – May and only limited rainfall in summer, resulting in a much more severe decrease in streamflow.

The effects of the drought only became fully visible in WY 2019/20 in the urban area – as seen by the lowest discharges in that year (compared to following years).

When plotting the double mass curves, the differences in cumulative amounts become even clearer between WY2019 and the following WY2020 (see below), with the effects of drought being visible in WY 2020 and also the imbalance between precipitation and evaporation in the rural catchment.

Section 4.3: Besides flow duration curves, I recommend adding double-mass curves (cumulative precipitation vs. cumulative specific discharge) for comparing the hydrological response of the two catchments during different years and at the seasonal scale (the focus could be on drought periods as well as on very wet months).
**We added double-mass curves to Figure 4.

Section 4.4: The characterization of the rainfall events should be anticipated in the results and merged to the first section of the results, in order to help understand the hydrographs and the flow duration curves.
**We restructured the section to present the seasonal distribution as well as characterization of rainfall events at the beginning of the result section (L 271- 321), followed then by the results of streamflow patterns and flow duration (L355-375).

Section 4.5: What is the sensitivity of young water fraction estimates on the sampling design for both rainfall (how many collectors were used?) and stream water? I wonder whether capturing the isotopic variability during flashy events in the urban catchment would have determined a different estimation. Furthermore, in this case, results on young water fractions and MTT may be due to a combination of factors, such as catchment area, geology and land use.
**For the collection of rainfall isotopes, generally only 1 collector is used. We did use two separate datasets of precipitation isotopes from Berlin Steglitz (for the urban catchment) and from the AWS in Hasenfelde (for the rural catchment).
Regarding sampling of stream water isotopes, we collected daily samples to insure continuity. This is a high resolution for stable isotopes in particular as sampled of longer periods (inter-annually). However, it is likely that sampling over the course of a rain event could give different estimates of young water contributions, which may be more damped when sampling just on daily basis, especially in the urban catchment where runoff responses can be relatively quick. From previous sampling, we know that there is minimal effect on young water estimates during flow peaks.
We agree, that estimates of young water fractions and MTT are likely due to a combination of factors related to land use and catchment area, and in the case of the urban catchment – the overwhelming influence of wastewater, which complicates the estimation of MTTs.

Table 2: Adjusted R2 are very low and RSE are very high for the urban catchment. Perhaps, these values should be considered carefully during the interpretation of the results.
**Yes, we agree these values need to be considered with care. They highlight the difficulties of these methods to estimate young water fractions in urban catchments with a strong influence of wastewater, as the isotopic variability is much more damped than in the rural stream,

resulting in higher RSE and lower R2. We included a caveat in the in the discussion section regarding interpretation.

Section 5.3 should be revised (please see the comment about the inter-comparison); based on this text and Figure 7, it seems that land use represents the main factor determining the different hydrological functioning of the two catchments during drought and wet periods.
**As mentioned also to Referee 1, we amended the schematic graphic in such a way that there is less focus on land use effects and more just on the aspects of different streamflow patterns under drought/wet periods in different anthropogenic environments. We also added amounts of young water estimates (in months) and flux amounts (in mm) to give a specific link to the results.

Figure S2: In late April of WY2020, there should be a rainfall event triggering the flash and marked discharge increase; however, there is only a rainfall pulse with a very low magnitude. Is this correct? Discharge in WY2020 seems to have a very different behaviour compared to the following years; is there a specific explanation?
**These distinct flashes in the urban streamflow regime are not necessarily associated with rainfall pulses but rather with streamflow management and opening of a weir upstream. This generally triggers such a marked streamflow response downstream, as is visible between late April and May 2020. This is usually done during drier periods – as was the case during the spring of 2020, to increase the flowrate and avoid the stream drying out. In this particular case the increased flow lasted for about 2,5 weeks before being reduced again (weir closed).

**Technical corrections**
Line 84: 'as part of' can be deleted.
**Deleted

Line 96: 'selected' instead of 'select'.
**Corrected

Line 100: 'in an integrated way' can be deleted.
**Deleted

Line 216: 'MTT' instead of 'MMT'.
**Corrected

Figure 2: For baseflow I see pink or purple lines, not red lines. Furthermore, I do not see the winter and summer events highlighted in red and green, respectively.
**Colors changed. Figure description amended.

Line 398: If the correlation is negative, r should be -0.49.
**Corrected

Line 434: '2018-19' instead of '208-19'.
**Corrected

Line 499: 'due to' repeated twice.
**Corrected

Line 500: Unclear 'lack of recharge. Hydromorphic conditions…'.
**Corrected

**Additional References:**

Bonneau, Jeremie, et al. "The impact of urbanization on subsurface flow paths–A paired-catchment isotopic study." *Journal of Hydrology* 561 (2018): 413-426.

von Freyberg, J., Allen, S. T., Seeger, S., Weiler, M., & Kirchner, J. W. (2018). Sensitivity of young water fractions to hydro-climatic forcing and landscape properties across 22 Swiss catchments.*Hydrology and Earth System Sciences*, *22*(7), 3841-3861.

Kleine L, Tetzlaff D, Smith A, Goldhammer T, Soulsby C. (2021) Using isotopes to understand landscape-scale connectivity in a groundwater-dominated, lowland catchment under drought conditions. *Hydrological Processes.* http://dx.doi.org/10.1002/hyp.14197

Kleine, L., Tetzlaff, D., Smith, A., Wang, H., & Soulsby, C. (2020). Using water stable isotopes to understand evaporation, moisture stress, and re-wetting in catchment forest and grassland soils of the summer drought of 2018. *Hydrology and Earth System Sciences*, *24*(7), 3737-3752.

Marx, C., Tetzlaff, D., Hinkelmann, R., & Soulsby, C. (2021). Isotope hydrology and water sources in a heavily urbanized stream.*Hydrological Processes*,*35*(10), e14377.

Marx, C., Tetzlaff, D., Hinkelmann, R., & Soulsby, C. (2023). Effects of 66 years of water management and hydroclimatic change on the urban hydrology and water quality of the Panke catchment, Berlin, Germany. *Science of the Total Environment*, *900*, 165764.

Tetzlaff D, Seibert J, Soulsby C. (2009) Inter-catchment comparison to assess the influence of topography and soils on catchment transit times in a geomorphic province; the Cairngorm Mountains, Scotland. *Hydrological Processes*, 23, 1874–1886.

Tetzlaff D, Seibert J, McGuire KJ, Laudon H, Burns DA, Dunn SM, Soulsby C. (2009) How does landscape structure influence catchment transit times across different geomorphic provinces? *Hydrological Processes* 23, 945–953

---

## Author Response (AR2)

Response to Editor Review:

Dear Editor,

Thanking for taking the time to provide the second review. We appreciate the comments and have addressed them accordingly.

In your response to #R1 you write: "The main novelty of this manuscript stems from the use of isotopes, as it gives context to understanding the different catchment responses."
This sentence should be included to the revised manuscript.
** We added this statement to the discussion section (L514-516).

L150 There are a few random references appearing next to the heading text

**Removed
3 Data and methods: #R1 raised some concern about a confusion here between seasonal patterns and response to storm events. A short mentioning why these two type of indicators are present here could easily resolve this confusion.

**We removed the sentence (L173-174) to avoid confusion. We only use the hourly precipitation data to look at storm events as well as hydrologic responses (i.e. flashiness, baseflow index) to these events.
L167 dot and space missing.

**Corrected
L174 there is an extra space here (October-May)

**Corrected
Fig. 2 feels very busy, and I am also worried if some detail could be lost in the final manuscript version. My recommendation would be either reducing the content of the figure, or moving its full version to the appendix and keep here only some zoomed-in time periods.
- I did not find any legend for the blue markers of the isotopes

**We moved the groundwater level plots to the supplementary material and added a reference to the figure in the manuscript. The grey grid lines have been removed to also reduce content. Otherwise, we would leave the figure as is – font size of legend has been increased.
- in general: the a)b)c)d) markings over the figures are inconsistent, here they are in bold, later just normal font

**Changed

L331 this sentence could use some figure references.

**Added reference

Fig. 3 here the letter b) is missing

**Added

in c) the in figure text is squeezed

**Changed

I find it difficult that the plots a) and b) have inconsistent labeblings. You should remove the arrows from b) and put there the Urban/Rural legend from a)

**Changed

also the color choice for the second Y axis is very light, a standard black labeling here would be perfectly fine

**Changed

general: be consistent with the ordering of the examples, if you always have the urban catchment first then the rural - in some figures it gets mixed up. In the caption of figure 3 I think the urban-rural is mixed up for c) and d)

**All figures now show first rural and then urban results

L337 is this the correct figure reference?

**Changed

Fig.4 typos in the caption

**Changed

In your response to #R1 you wrote:

"As a general interpretation: the closer the values are together, the less variable they are - meaning a more constant and similar water source is present in a stream, while points spread larger apart indicate greater variability in the source water contributions and seasonal variability" Including these sentences to the main text would strongly help the understandability of this section.

**Included sentence in main text (L352-354)

L376: are water ages and young water fractions shown in fig 5. If yes, this should be mentioned in the caption in some ways.

**Only young water fractions are shown. Explanation added to the figure caption.

Fig.5 here the rural catchment comes before the urban which is very confusing.

What is the gray line?

**Grey line indicates the sinusoidal cycle for precipitation – added explanation to figure

caption.

Fig. 6: in some captions you use abbreviations only (here TPLR), sometimes with full words and the abbreviations. From a reader perspective the second solution would be much easier to follow.

**Added full term in the caption

Also please check if all abbreviations are introduced in the text properly. For example WY is not. As the SI is targeting a broader audience even some trivial conventional abbreviations should be introduced.

**Added explanation for WY abbreviation (L164) and checked all other abbreviations

In the revised manuscript I found the discussion section more fitting to the whole text.

L454 there is a problem with a reference here

**Corrected

L486 the abbreviation $F\_yw$ is just $F_{yw}$ here.

**Corrected

L560: Copernicus journals prefer to have a full reference for these data websites, for the Wasserportal a working way of reference would be:

**Corrected

The data can be accessed at: Public discharge data is available from: https://wasserportal.berlin.de/stationen_start.php (SenUVK, 2023) and the reference to it: SenUVK. (2023). Wasserportal Berlin. https://wasserportal.berlin.de/stationen_start.php

Response to Reviewer #1:

Thank you to this reviewer for taking the time to review again our revised manuscript. We appreciate the attention to detail and final comments. We made the necessary changes in the manuscript.

Line 85: 'catchments' is missing between 'contrasting' and 'over'.
**Changed

Line 150: There is a list of three references to be deleted after the section title.
**Changed

Lines 157-158: There is a missing verb in this sentence.
**Corrected

Lines 252-253: Please use normalized discharge values.

**Corrected

Figure 2: Please increase font size in the legends.

**Changed

Lines 355-356: You should refer to the LMWL, particularly if there is a marked difference in the slopes of GMWL and LMWL. Anyway, if most of the samples plots below the line, there is evidence of evaporation.

**Changed

Lines 356-358: Is the reference to Figure 4b correct? I think this observation should be based on Figure 5.

**Changed

Lines 377 and 379-380: Please write p<0.001 instead of p<2.2e-16.

**Changed

Lines 384-386: If the correlations are negative, r must be negative. Besides this, it seems strange that young water fractions decrease with an increase in annual discharge.

**Changed